# Harnessing Non-Adversarial Robustness in Large Language Models

**Qinghua Zhou** [1 2]  **Ellina Aleshina** [3]  **Andrey Lovyagin** [3]  **Oleg Somov** [4 5]  **Mikhail Seleznyov** [4 3]
**Alexander Panchenko** [4 3]  **Ivan Oseledets** [4]  **Elena Tutubalina** [4]  **Ivan Y. Tyukin** [3 1 2]

## Abstract

The work presents an approach for addressing the challenge of robustness in Large Language Models (LLMs) to alterations and potential errors caused by semantically similar but textually different prompts. Recent works have shown that these kinds of prompt variations can significantly impact the performance of LLMs on tasks. The central question is: can LLMs' robustness to semantically-neutral prompt alterations be acquired without expensive retraining of the entire model? We address this question both theoretically and through experiments. Our theoretical analysis reveals a crucial factor impacting model robustness – a systematic expected shift or perturbation-induced bias in neural network module outputs. Motivated by this analysis, we show that robustness can be achieved via a simple fine-tuning process: debiasing for robustness. We identify conditions when debiasing helps and when it does not, and demonstrate, through both theory and extensive experiments, that debiasing for robustness may indeed be a quick and efficient tool to enhance robustness and provide certification against random prompt perturbations.

## 1. Introduction

Robustness of LLMs with respect to prompt perturbations is a standard requirements for modern LLMs. A large body of work on the topic of ensuring robustness has already been developed to date (see Kumar & Mishra (2025) for an overview). Most of these approaches are based on targeted and appropriately designed training algorithms (Yu et al., 2024; Anthony & Bartlett, 2009; Kim et al., 2021),

similar to the standard approaches in other areas such as computer vision (Xie et al., 2020). Whilst these methods broadly apply to a large number of models, the sheer size of LLMs prevents broad application of these methods due to the volumes of computational resources and training data needed to implement these measures.

Alternatives such as post-training certificates provided by, e.g., randomized smoothing (Cohen et al., 2019; Zeng et al., 2023) require multiple inference runs per prompt and may be computationally expensive. Moreover, they may not warrant high accuracy as their performance, by the very definition of the randomized smoothing, is the expected performance of the original model on perturbed inputs. If the latter is poor, then so is that of the stabilized model.

Remedies such as self-denoising (Zhang et al., 2023) demonstrate potential to improve accuracy in experiments, but they retain high computational burden needed to implement these tools. Moreover, the application of all these methods is typically constrained to classification tasks and may not be immediately applicable to other tasks.

*The question, therefore, is: if and when improvements of LLMs' performance can be achieved via small and possibly task-dependent modifications of the model, without the need for significant computational budgets and supervisory ground truth information?*

In this work, we propose a novel holistic approach to improve both robustness and accuracy of LLMs, relative to their original performance on perturbed data. The approach is grounded in the insights stemming from the theoretical analysis of certified robustness of LLMs that accounts for major traditional determinants of robustness, such as Lipschitz constants and margins, as well as relevant statistical features of models operating in perturbed environments. The approach, in principle, is not confined to classification tasks and does not require full model retraining, albeit it enables the inclusion of fine-tuning via LoRA (Hu et al., 2022), in-context learning (Dong et al., 2024), or other standard methods into the proposed workflow.

At the heart of our approach is the intuition that nonlinearities inherent in LLMs may be exploited to navigate around the robustness-accuracy-retraining dilemma with-

---

[1]International Joint Laboratory of AI for Industry, QUST, Qingdao, China [2]King's College London, London, UK [3]Applied AI Institute, Moscow, Russia [4]AXXX, Moscow, Russia [5]MIRIAI, Moscow, Russia. Correspondence to: Ivan Tyukin <ivan.tyukin@kcl.ac.uk>.

*Proceedings of the 43rd International Conference on Machine Learning*, Seoul, South Korea. PMLR 306, 2026. Copyright 2026 by the author(s).

out spending excessive computational resources to achieve appropriate levels of robustness (in contrast to Lipschitz or large margin-constrained training (Pethick et al., 2025; Huang et al., 2018)). The advantage gained from a somewhat more nuanced accounting for model nonlinearities in robustness certificates, as opposed to nonlinearity-agnostic certificates (randomized smoothing), is illustrated in Figure 1. In Figure 1, $M(x)$ stands for the feature vector corresponding to some input data/prompt $x$. Naive intuition defines a robustness margin for $x$ as the distance from $M(x)$ to a decision boundary. However, if we consider the expectation of $M(x + \delta x)$, where $\delta x$ is a random perturbation, then the value of $E_{\delta x}[M(x + \delta x)]$ may be far away from $M(x)$ (see Figure 8 in Appendix A for the evidence of this occurring in LLMs). Hence "typical" behavior of models without and under perturbations could be radically different. This includes distances to decision boundaries and hence robustness. Therefore, taming the change in expected performance of perturbed models could be associated with an appropriate compensation for the shifts induced by both the perturbations themselves and nonlinearities in LLMs.

In the setting of harnessing nominal, non-adversarial robustness, the main contributions of our work are:

- We identify a critical robustness factor for LLMs – a shift in expected statistical properties of the model's latent features arising due to external perturbations.

- We provide robustness certificates linking major known determinants of robustness, such as sparsity and Lipschitz constants, into unified expressions for both simple and complex noise models.

- Guided by the theoretical analysis, we present simple and efficient debiasing methods to remedy negative impact of random perturbations in the absence of task supervisory information, and show these methods also improve certification.

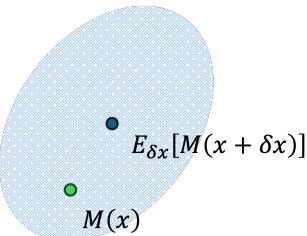

Figure 1. Geometric intuition behind *perturbation-induced bias*: there is no guarantee that the expectation of perturbed inputs representations' center will coincide with the unperturbed input, i.e. $E_{\delta x}[M(x + \delta x)] \neq M(x)$.

The paper is organized as follows. In Section 2 we review relevant literature and formulate specific research questions,

Section 3 describes the overall theoretical framework. Sections 4, 5 present new algorithms and their empirical verification. In Section 6 we provide discussions in a broader context and highlight limitations. Section 7 concludes the paper.

## 2. Prior work

Ensuring robustness to perturbations of input data has been the topic of major effort in the past. The most relevant approaches for the current work are discussed below.

*Large margin classifiers* (Sokolić et al., 2017; Anthony & Bartlett, 2009) are widely used to create both accurate and robust models. A related approach is *adversarial training* (Xhonneux et al., 2024) whereby robustness is imposed via appropriately chosen training loss and model training protocols. These approaches, however, require supervisory ground truth data and availability of major computational resources for training, a task which is substantially different from the one we are considering here.

*Spectral normalization*, proposed in Miyato et al. (2018), enables controlling the spectral norm of weight matrices in the model. This has the potential to ensure that the overall Lipschitz constant of the model is kept small, typically less than one. It has been shown in Bartlett et al. (2017) (Theorem 1.1) that small spectral norms imply better robustness guarantees. Yet, similar to large margin classifiers, the method assumes complete model retraining – the pathway we are striving to avoid. Moreover, some mappings within LLMs may not necessarily be Lipschitz everywhere (Dasoulas et al., 2021).

*Batch calibration*, introduced in Zhou et al. (2023), is a post-training technique designed to control contextual bias from a batched prompt. The method, in essence, estimates empirical contextual bias from a batch of prompts and then applies the adjustment during inference. Seleznyov et al. (2025) showed that the method can, in principle, be used to address robustness to semantically-neutral prompt perturbations. It does not, however, yield robustness certificates.

In our paper, we reveal that a similar effect can be achieved through direct modifications of model parameters. This (i) eliminates the need for external corrections at the stage of class label predictions and (ii) creates the potential for applications other than mere classification tasks. Moreover, our approach leads to testable robustness certificates.

*Randomized smoothing* (Cohen et al., 2019) and its LLM-adapted versions with prompt masking (Zeng et al., 2023) and template ensembling (Voronov et al., 2024) are popular post-hoc approaches to warrant worst-case robustness. They are applicable to generic models, regardless of their architecture. Yet, whilst delivering robustness guarantees, these

methods are computationally expensive as they assume multiple inference passes per single prompt. Moreover, they apply largely to classification tasks. Our aim is to remove these obstacles through inexpensive parameter tuning.

*Low-Rank adapters (LoRA) and fine-tuning.* A recent empirical study (Seleznyov et al., 2025) thoroughly assessed several methods for increasing robustness of LLMs to semantically-neutral perturbations across 52 tasks from the Natural Instructions dataset (Wang et al., 2022). This study looked at a host of robustness-enhancing approaches such as few-shot learning, LoRA, sensitivity-aware decoding (Lu et al., 2024), and template ensembling (Voronov et al., 2024). The authors observed that naive application of LoRA, including LoRA variants with prompt augmentation, did not result in consistent improvements in robustness. Moreover, in cases when LoRA adapters or other fine-tuning improve robustness, they still require access to supervisory ground truth information about expected tokens or labels and do not produce *a priori* computable robustness certificates.

This raises the following research questions:

(i) What are the origins behind the widely reported LLMs' apparent fragility in the presence of non-adversarial semantically-neutral perturbations (see also our own experiments in Appendix, Table 6 evidencing this behavior)?

(ii) Can this robustness drop be mitigated by appropriate interventions inside the model and in the absence of any supervisory information?

(iii) Can we certify robustness at both example and population levels without accessing ground truth responses and additional inference runs?

In the next section, we present a theoretical framework that could be used to address these questions. The theory enables producing robustness certificates linking major determinants of robustness, such as sparsity and Lipschitz constants (if defined), into a single expression. Most importantly, our analysis identifies that one of the major factors behind the loss of robustness is a shift in expected statistical properties of the model's latent features (see Remarks B.2, B.6 and Theorems B.1, B.5). Guided by this, in Sections 3.3-3.4 and Appendix C we propose a robustness mitigation approach supported by relevant robustness guarantees (see Theorem C.2 and its generalization in Appendices D-F).

## 3. Theoretical analysis

In what follows, we first consider a generic setting in which the main object of interest is mappings $M : \mathbb{R}^n \to \mathbb{R}^d$ modeling internal transformation of data in LLMs.

### 3.1. Data model

Given a module $M$, we assume that the input data is the following multi-set:

$$S = \{x_1, \ldots, x_m\}, \; x_i \in \mathbb{R}^n, \; i = 1, \ldots, m$$

Elements of the multi-set $S$ can be immediate outputs of the LLM's tokenizer, outputs of model's intermediate layers, or outputs from the penultimate layer of the model. The values $M(x_i)$, $i = 1, \ldots, m$, could, in turn, represent outputs of any relevant module as functions of $x_i$, including multi-layer perceptron (MLP), self-attention, their stacked combinations, or final logits.

In order to keep the theory as general as possible, we do not wish to impose any further assumptions on the nature of the input data $x_i$. They may or *may not* be sampled from some distribution or be associated with supervisory information. In cases where $x_i$ turn out to be samples from a distribution, no standard i.i.d. assumptions are imposed on these samples *a priori*.

### 3.2. Model of perturbations

Perturbations to input data $x \in S$ are assumed to be random variables $\delta x$, whose components are independent but not necessarily identically distributed. More formally, for an original data vector $x \in S$, a perturbed data vector, $\hat{x}$, is

$$\hat{x} = x + \delta x,$$

where

$$\delta x = (\delta x_1, \ldots, \delta x_n)$$

is a vector modeling perturbations. To simplify the presentation and reveal the gist of the phenomenon, one may assume that $\delta x_i$ are independent and bounded random variables $\delta x_i$:

$$\delta x_i \in [-c, c], \; c \in \mathbb{R}, \; c > 0.$$

Yet, as we show in Appendix E, the theory can be generalized to settings in which both the boundedness and independency assumptions may be dropped.

### 3.3. Perturbation-induced bias

Let $M_i$ be the $i$-th component of the map $M$. Consider the function

$$f_i(x, \delta x) = M_i(x + \delta x) - M_i(x),$$

and let $\mathbb{E}_{\delta x}\left[f_i(x, \delta x)\right]$ be the expectation of $f_i(x, \delta x)$ at $x$. If $\mathbb{E}_{\delta x}\left[f_i(x, \delta x)\right] = 0$, expected or typical response $\mathbb{E}_{\delta x}[M_i(x + \delta x)]$ of the perturbed feature coincides with the model's response $M_i(x)$ to the clean input $x$. As we show in Figure 8 in the Appendix, for a large proportion of features in an LLM this assumption may not hold true. In such cases, perturbations induce *bias* $\mathbb{E}_{\delta x}\left[f_i(x, \delta x)\right]$ at $x$.

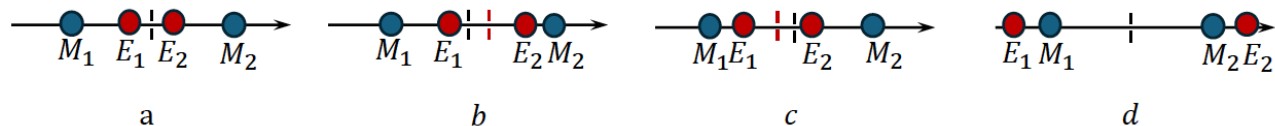

*Figure 2.* Examples of possible outcomes due to perturbation-induced bias. $M_1$ and $M_2$ denote $M_i(x_1)$ and $M_i(x_2)$, $E_1$ and $E_2$ denote $\mathbb{E}_{\delta x}[M_i(x_1 + \delta x)]$ and $\mathbb{E}_{\delta x}[M_i(x_2 + \delta x)]$. Black dashed line indicates decision threshold. *Panel a:* Overall reduction of robustness, and debiasing is not effective. *Panels b,c:* Overall reduction of robustness, but reduction in robustness radius can be compensated for by debiasing; this effectively moves the decision threshold (shown by red dashed line). *Panel d:* Overall enhancement of robustness due to random perturbations. Debiasing may not be needed here (see tasks in Table 7 with positive effect of perturbation on performance for examples).

Note that perturbation-induced bias $\mathbb{E}_{\delta x}[f_i(x, \delta x)]$ differs from the broader concept of distributional shift (Bansak et al., 2024) in that it specifically captures expected impact of additive perturbations regardless of the nature of the training data. It is also different from internal covariate shift (Ioffe & Szegedy, 2015) as it may and will likely occur during and after training.

The value of perturbation-induced bias depends on $x$. Given that the set $S$ is finite, one can assume the availability of an upper bound $C_i$:

$$\sup_{x \in S} |\mathbb{E}_{\delta x}[f_i(x, \delta x)]| \leq C_i, \ i = 1, \ldots, d.$$

It turns out (see Appendix A for a detailed theoretical analysis for this setting) that perturbation-induced bias is a major factor contributing to the loss of robustness in the presence of random perturbations.

Indeed, suppose that $M_i : \mathbb{R}^n \to \mathbb{R}$ for $i = 1, ..., d$ is $\gamma_i$-Lipschitz, i.e.,

$$|M_i(x) - M_i(y)| \leq \gamma_i \|x - y\|,$$

and the model's performance is deemed unaffected by perturbations as long as

$$|f_i(x, \delta x)| \leq \epsilon_i \text{ for all } x \in S.$$

The value of $\epsilon_i$ could be viewed as the model's assured robustness radius, where a smaller radius indicates better consistency in $M_i$ outputs to the expected response. In this case, from Theorem B.1 and Remark B.2 in Appendix A, if $1 - \phi_i, \phi_i \in (0, 1)$ is the desired confidence of the above event,

$$\epsilon_i(\phi_i) = C_i + c\gamma_i \sqrt{2n \ln\left(\frac{2m}{\phi_i}\right)}. \tag{1}$$

A similar expression that applies to non-Lipschitz mappings $M_i$ is provided in Appendix B (see Theorem B.5 and Remark B.6):

$$\epsilon_i(\phi_i) = C_i + \sqrt{\frac{V_i m}{\phi_i}}, \ \mathrm{Var}[f_i(x, \delta x)] \leq V_i. \tag{2}$$

This means that working with models whose inherent robustness radius is large requires feature debiasing – that is, reducing the values of $C_i$.

### 3.4. Mitigating against perturbation-induced bias

The presence and the revealed impact of perturbation-induced bias on model's performance motivate a simple and computationally efficient mitigation strategy.

The strategy entails replacing the original module $M_i$ with a bias-corrected module $M_i^*$:

$$M_i^*(x) = M_i(x) - b_i, \ b_i \in \mathbb{R}, \tag{3}$$

where $b_i$ is the bias-correcting term. The rationale behind this choice is the observation that

$$\inf_{b_i} \sup_{x \in S} |\mathbb{E}_{\delta x}[f_i(x, \delta x)] - b_i| \\ \leq \sup_{x \in S} |\mathbb{E}_{\delta x}[f_i(x, \delta x)]|.$$

Hence appropriate tuning of $b_i$ enables reducing the values of $C_i$ (see Theorem C.2 for further details).

Does the reduction of bias always improve the performance of models in tasks? To gain an insight, consider a binary classification task, and let $M_i(x)$ be a decision variable determining model's classification outcomes: the model assigns $x$ to class 1 if $M_i(x) > 0$ and class 2 otherwise. Suppose that $M_i(x_1) < 0$ and $M_i(x_2) > 0$. In theory, perturbation-induced bias may lead to different outcomes (see Figure 2). Panel $a$ shows a setting when perturbation-induced bias reduces robustness but debiasing does not help. Panels $b, c$ visualize scenarios whereby debiasing may improve robustness. Panel $d$ depicts a rather unusual but nevertheless plausible setting whereby perturbations "help" to increase robustness. Debiasing, however, does not improve robustness (similar to the case shown in Panel $a$). Remarkably, all four cases were observed in experiments (Table 7).

**Takeaway:** Classical determinants of robustness such as Lipschitz constants $\gamma_i$, robustness radius $\epsilon_i$, and sparsity (the number of relevant attributes $n$) have long been considered as major factors contributing to model stability and robustness to perturbations. Yet, as this analysis shows,

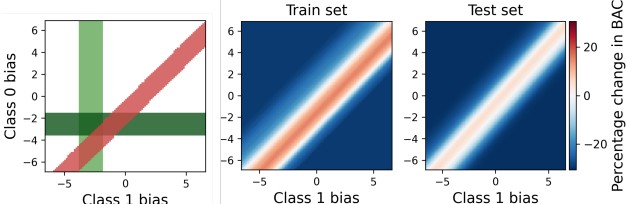

*Figure 3.* Impact of (constant) bias $b_i$, for the bias-corrected module $M_i^*(x) = M_i(x) - b_i$ on a random Natural Instructions task selected in Section 4. Right figures show BAC of perturbed prompts; left figure show top 15-percentile improvement regions for performance (red) and $\epsilon_i(\phi)$ for two neurons (green). There exists an overlapping region of optimal intervention.

these factors (e.g., small Lipschitz constants $\gamma_i$ or small variance if $M_i$ is not Lipschitz) do not, on their own, warrant robustness. In fact, when the value of bias $C_i$ is large, controlling robustness by mere reduction of Lipschitz constants or variance by making them arbitrarily small becomes infeasible. Therefore, robustness debiasing via direct interventions inside neural network modules, as in (3), is a simple and computationally efficient way to mitigate performance deterioration in the presence of perturbations.

Detailed theoretical analysis of both the simplified and general settings, including theoretical robustness certificates and the effect of bias correction, is provided in Appendices A-F. In the next sections, guided and motivated by the theory, we present new methods for harnessing non-adversarial robustness and the corresponding experimental setup.

## 4. Experimental setup

Given the focus of our work, the experimental setup below concerns prompts' variations that are semantically equivalent to a human reader but differ in surface textual forms.

For clean and unperturbed prompts, we consider tasks from Natural Instructions (Wang et al., 2022), following the task selection protocol of Seleznyov et al. (2025) and applying additional task filters to focus on the problem of interest – robustness to non-adversarial perturbations and methods for improvement. These additional task filters are: (1) a > 1% drop in performance due to perturbations, (2) balanced accuracy (BAC) $\geq 70\%$ on original examples from the benchmark (*clean* examples). Names and descriptions of all selected tasks can be found in Appendix G.1.

We work with raw logit outputs of the logit layer on binary and multi-class classification tasks where the first generated token determines the predicted label/answer. In this scenario, each neuron used in the classification represents a component map $M_i(x)$, providing us with a simple and explainable setup.

Two common types of semantically-neutral perturbations

are considered: format perturbations and text perturbations. Format perturbations follow Seleznyov et al. (2025) and modify the structural elements of the prompt template, while keeping the core prompt unperturbed. Text perturbations change prompts through zero-width Unicode-confusables, markup wrappers, and keyboard typos. See Figure 9 in the Appendix for examples.

We evaluate on base variants of three open-weight models: Qwen-3-8B, Llama-3.1-8B, and Olmo-3-7B. Applying the task filters yields 9 unique tasks with 19 task-model pairs for format perturbations and 16 task-model pairs for text perturbations. We will refer to these as *target tasks*. For each task, we sample 200 examples and 200 perturbations per example for each perturbation type. The dataset is split across three sets, 50% training, 35% testing, and 15% hold-out for population-level certificates. For text perturbations, random perturbations were applied to examples in each set. For format perturbations, a separate set of formats was used for each set (Appendix G.2).

## 5. Debiasing for robustness

### 5.1. Logits debiasing

Theoretical analysis in Section 3 and Appendix A reveals the potential emergence and impact of systematic expected shifts in module outputs under perturbation. Hence, to improve robustness, we may intervene on $M_i(x)$ with principled approaches: (1) input-independent debiasing, and (2) input-dependent debiasing.

#### 5.1.1. INPUT-INDEPENDENT DEBIASING

**Method:** A simple intervention approach is to add a bias $b_i$ to neurons to calibrate the logits, replacing $M_i(x)$ with $M_i^*(x) = M_i(x) - b_i$. Here $b_i$ is constant and independent of input $x$. As shown in Theorem C.2, we can define a new perturbation-induced bias bound $C_i^*(b_i)$. Under the practical assumption that the model is performant on clean prompts, we can search for $b_i$ minimizing $C_i^*(b_i)$, "debiasing" the logit outputs.

For this method, we limit ourselves to binary classification tasks. Figure 3 (left) presents the change in perturbed prompt accuracy for different values of $b_i \in [-C_i, C_i]$, where we can observe that there is an optimal offset where accuracy can be improved through the introduction of the bias. For input-independent debiasing, we search for overlapping regions of improvement above the 90th percentile for robustness radii (1), (2), and BAC on the training set of examples and perturbations. This overlapping region is shown in Figure 3 (right).

**Metrics:** To examine the effect of perturbations on models' performance, we measure the difference in BAC between

*Table 1.* Example input-independent and input-dependent debiasing for a random binary Task 065 with format perturbations (top section) and text perturbations (bottom section). Measurements based on $m = 70$, $\phi = 0.05$. For input-independent debiasing, rows with no values show no overlapping region of improvement for both perturbed prompt BAC and $\epsilon_i(\phi)$ in the train set. See Sections 5.1.1-5.1.2 for description of the measured metrics. Here red and blue indicate decrease and increase in performance/robustness due to debiasing, respectively. See Tables 5-6 in the Appendix for results on more tasks.

| Model | Input-independent | | | | | Input-dependent | | |
| | $\delta_{\text{drop}}$ | $\delta_{\text{clean}}$ | $\delta_{\text{pert}}$ | $p_{\epsilon L}$ | $p_{\epsilon V}$ | $\delta_{\text{clean}}$ | $\delta_{\text{pert}}$ | $p_{\epsilon V}$ |
|---|---|---|---|---|---|---|---|---|
| Llama | -20.9 | -10.1 | 11.1 | 75.9 | 53.1 | -1.9 | 16.9 | 81.9 |
| Qwen | -5.7 | – | – | – | – | -2.3 | 5.4 | 51.5 |
| Olmo | -13.4 | – | – | – | – | -9.6 | -1.3 | 13.9 |
| Qwen | -6.1 | – | – | – | – | -9.5 | 12.6 | 90.3 |
| Llama | -16.7 | – | – | – | – | -12.9 | 5.7 | -45.9 |
| Olmo | -21.8 | 2.7 | 7.8 | 35.5 | -11.6 | -1.1 | 4.9 | 52.8 |

*Table 2.* Measuring impact of input-dependent debiasing across target tasks for models with format perturbations (top section) and text perturbations (bottom section). See Sections 5.1.1-5.1.2 for description of measured metrics. See Table 8 in the Appendix for results on each task and Table 9 for confidence intervals.

| | Llama 8B | | Olmo 8B | | Qwen 7B | |
| | Mean | Std | Mean | Std | Mean | Std |
|---|---|---|---|---|---|---|
| $p_{\text{damage}}$ | 26.2 | 21.6 | 16.7 | 12.6 | 17.9 | 12.8 |
| $p_{\text{recover}}$ | 20.1 | 96.6 | 67.8 | 42.6 | 93.4 | 6.1 |
| $p_{\text{clean}}$ | -4.9 | 5.1 | -3.7 | 7.2 | -2.9 | 3.6 |
| $p_{\text{combined}}$ | 28.8 | 33.6 | 16.3 | 24.3 | 22.6 | 21.1 |
| $p_{\epsilon V}$ | 83.8 | 9.1 | 66.1 | 23.1 | 76.5 | 17.9 |
| $p_{\text{damage}}$ | 21.3 | 3.6 | 26.9 | 8.5 | 2.8 | 3.9 |
| $p_{\text{recover}}$ | 66.4 | 30.6 | 51.1 | 31.1 | 55.2 | 67.1 |
| $p_{\text{clean}}$ | -16.4 | 18.5 | -12.8 | 11.4 | -1.1 | 0.9 |
| $p_{\text{combined}}$ | 16 | 7 | 18.1 | 10.8 | 2.6 | 3 |
| $p_{\epsilon V}$ | 74.3 | 19.4 | 46.2 | 43.6 | 78.3 | 22.1 |

clean and perturbed prompts: $\delta_{\text{drop}}$. It quantifies the performance gap we want to recover with debiasing. To examine the effect of debiasing on performance, we measure the change in BAC (relative to the unaltered model) on clean and perturbed prompts with a bias selection mechanism ($\delta_{\text{clean}}$ and $\delta_{\text{pert}}$, respectively). To examine the effect of debiasing on robustness, we measure the mean percentage decrease in robustness radii across neurons, $p_{\epsilon L}$ and $p_{\epsilon V}$ for robustness radii (1), (2).

**Results:** We examine multiple binary, balanced classification tasks from the set of target tasks. An example of input-independent debiasing across models and perturbation types for a randomly chosen single task is shown in Table 1 (results on other tasks are shown in Table 5 in the Appendix). These results illustrate the impact of $b_i$ on both $\delta_{\text{pert}}$ and $\epsilon_i(\phi)$. We note that debiasing in general results in improvements in $\delta_{\text{pert}}$ and contraction of at least one of the two robustness radii, but it often comes at the cost of performance on clean examples (negative $\delta_{\text{clean}}$). Moreover, optimal regions may not exist for all models and tasks.

**Takeaway**: An optimal overlap region between robustness and performance can exist for certain models and tasks, and input-independent debiasing may constitute a simple yet computationally efficient first-stop tool for improving robustness to semantically-neutral perturbations . However, input-independent debiasing is inherently constrained: only a few constants $b_i$ control performance. In the next section, we examine input-dependent bias $b_i(x)$ and provide an input-dependent method for debiasing.

### 5.1.2. INPUT-DEPENDENT DEBIASING

**Method**: To add flexibility, we consider the following input-dependent variant of the corrected module:

$$M_i^*(x) = M_i(x) - b_i(x)$$

The rationale is that by optimizing the value of

$$\sup_{x \in S} |\mathbb{E}_{\delta x} f_i(x, \delta x) - b_i(x + \delta x)|,$$

over $b_i \in \mathcal{F}$, where $\mathcal{F}$ is a class of functions $\mathbb{R}^n \to \mathbb{R}$ containing all constant functions, one can potentially achieve a more efficient compensation of $C_i$ and $V_i$. Here we define

$$b_i(x) = w_i \cdot \psi(x) + \beta_i$$

where $\psi(x) = \left[x^\top, 1\right]^\top \in \mathbb{R}^{(n+1)}$ and closed-form solutions for $w_i$ and $\beta_i$ are found via linear ridge regression (more details in Appendix G.3).

**Metrics**: We examine metrics introduced in Section 5.1.1, with the exception of $p_{\epsilon L}$, which is omitted since it requires Lipschitz regularization on the linear ridge regression. We also introduce other parameters aiding evaluation across target tasks. $p_{\text{damage}}$ measures the the percentage drop in BAC when perturbations are introduced relative to clean examples' BAC; $p_{\text{recover}}$ measures the percentage of $p_{\text{damage}}$ recovered by debiasing; $p_{\text{clean}}$ measures the percentage change of BAC on clean examples due to debiasing; $p_{\text{combined}}$ measures the percentage improvement over combined clean and perturbed examples BAC due to debiasing. In addition, we measure task-level confidence intervals (95% CI w.r.t. mean) with bias-corrected accelerated bootstrapping with 10k resamples.

**Results**: Examining the same example task in Table 1, we observe high $p_{\epsilon V}$ indicating improvement to the robustness radius. Table 6 in the Appendix shows results for all models, target tasks, and perturbation types. For most models and perturbations we observed improvement in $\delta_{\text{pert}}$, albeit at some cost to $\delta_{\text{clean}}$.

Table 2 summarizes performance measures for input-dependent debiasing across target tasks. For individual tasks and models, $p_{\text{damage}}$ ranges from $-0.5\%$ to $62.2\%$ (see results for each task in Table 8 in the Appendix). Combined

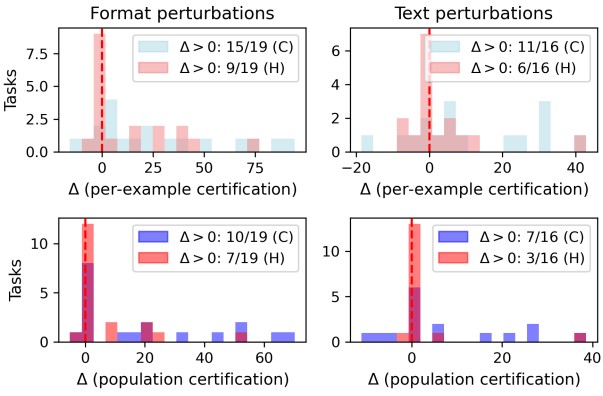

*Figure 4.* Effect of intervention on per-example and population-level certification. The histograms account for target tasks across models; in the legends we show the fraction of tasks with $\Delta > 0$.

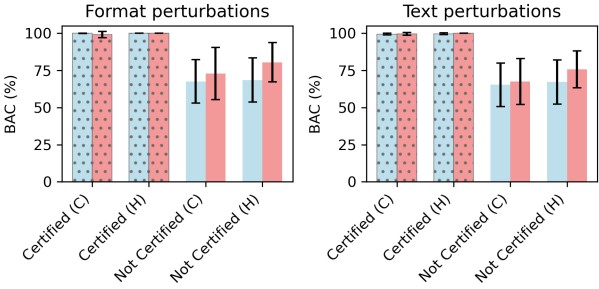

*Figure 5.* Empirical verification of per-example certification. Blue and red columns correspond to before and after debiasing, respectively. See Table 11 in the Appendix for detailed results.

with the results from Table 2, we observe overall substantial performance degradation. Negative $p_{\text{damage}}$ indicates performance improvement due to perturbations, corresponding to case $d$ in Figure 2. The $p_{\text{recover}}$ metric shows that input-dependent debiasing recovers a significant portion of the perturbation-induced damage in most cases, though at the cost of reduced clean-example performance (as shown by generally negative $p_{\text{clean}}$). This is further supported with confidence intervals in Table 9 in the Appendix. When perturbations occur frequently, debiasing becomes favorable, with positive $p_{\text{combined}}$ for most tasks. We also observe considerable improvement of the robustness radius, as expected.

**Takeaway**: Input-dependent debiasing provides an efficient and broadly applicable correction mechanism, recovering a significant portion of performance lost to perturbations while simultaneously improving robustness radii. The trade-off is a modest reduction in clean-example performance, but the net combined improvement remains positive across the majority of tasks, models and perturbation types.

## 5.2. Robustness certification

The component-wise stability setting in Sections 3.1-3.4 (and Appendices A-C) can be generalized to *margin stability* certification on task performance (see Appendices D-F). This combines performance and robustness under a single framework, enhancing the practical usability of robustness certification. We note, however, that whilst component-wise certification stemming from Sections 3.1-3.4 and Appendices A-C does not require ground truth data, margin certification typically does so in multi-class tasks.

**Method**: Based on Theorem F.1 and Remark F.3, a per-example certificate can be estimated for each clean example $(x, y)$ if the margin $\rho(x, y)$ exceeds a threshold determined by the expected margin shift under a reference (uniform) distribution plus a concentration term. Once certified, performance on perturbed variants of the example is guaranteed up to a failure ratio. The per-example level certificate requires access to the clean example. Based on Remark F.4 with appropriate assumptions, we can generalize to population-level certification; examples drawn from the same data distribution will have guaranteed performance on perturbed examples without requiring clean examples.

**Metrics:** Here we move from certification of robustness radii to certification of distances to decision boundaries. For per-example certificates, we predict the percentage of certified examples using Chebyshev ($p_C$) and Hoeffding's ($p_H$) inequalities. Under population certification assumptions, $P_C$ and $P_H$ provide population-level performance guarantees on perturbed examples. The primed variants $p'_C$, $p'_H$, $P'_C$ and $P'_H$ correspond to values after input-dependent debiasing. $\Delta$ is the difference in metrics after debiasing. The same task-level confidence intervals measured for debiasing metrics are also measured for certification metrics.

**Results:** Table 3 shows that base models can be certified for certain tasks, although to a limited extent. Figure 4 and Table 3 show an overall positive impact of debiasing on certification: a substantially higher percentage of examples are certified at the per-example level, and stronger population-level guarantees. This is further confirmed with confidence intervals in Table 14 in the Appendix. For some tasks (as shown in Table 10 and Table 12), debiasing drastically improves certification rates. Figure 5 verifies per-example level certification both before and after debiasing, showing that perturbed variants of certified clean examples achieve near 100% BAC (exceeding the failure ratio $\phi = 0.1$), with performance differing significantly between certified and uncertified prompts.

**Takeaway**: Debiasing can improve certification at two levels simultaneously: it increases the *proportion of margin-certifiable certifiable examples*, which translates to stronger *population-level* robustness guarantees (as shown in Ta-

*Table 3.* Mean certification metrics at both the per-example and population levels with/without input-dependent debiasing (with $\phi = 0.1$ and $\psi = 0.05$) for target tasks with format perturbations (top section) and text perturbations (bottom section). Primed metrics are measured after debiasing. See Section 5.2 for description of measured metrics. See Tables 10, 12 and 14 in the Appendix for detailed results.

| | Per-example | | | | Population | | | |
|---|---|---|---|---|---|---|---|---|
| **Model** | $p_C$ | $p_H$ | $p_C'$ | $p_H'$ | $P_C$ | $P_H$ | $P_C'$ | $P_H'$ |
| Llama 8B | 7.9 | 0 | 17.4 | 6.9 | 0.4 | 0 | 8.4 | 1.4 |
| Olmo 7B | 8.9 | 3.9 | 36.4 | 17.0 | 3.7 | 1.5 | 22.9 | 8.0 |
| Qwen 8B | 9.4 | 0 | 55.7 | 27.4 | 2.0 | 0 | 37.9 | 15.1 |
| Llama 8B | 20.7 | 4.8 | 29.5 | 6.2 | 9.2 | 0.4 | 15.8 | 0.1 |
| Olmo 7B | 8.8 | 2.7 | 13.5 | 1.6 | 3.0 | 0 | 4.6 | 0 |
| Qwen 8B | 32.4 | 12.4 | 63.8 | 31.0 | 19.0 | 1.6 | 42.8 | 15.1 |

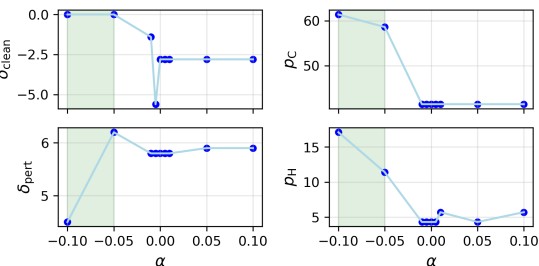

*Figure 6.* Effect of Gram penalty on input-dependent debiasing for Task 065. $\alpha$ is the parameter controlling the magnitude of the Gram penalty, enabling control over the clean-perturbed trade-off. Green region indicate $\alpha$ ranges where $\delta_{\text{clean}} \geq 0$. More examples demonstrating this effect can be found in Table 16 in the Appendix.

ble 3). Strikingly, this is consistent with what the theory predicts (see Figure 2, panels b and c, where debiasing effectively increases margins). Although the guarantees remain conservative, this indicates that reducing perturbation-induced bias and stabilizing margin statistics is beneficial not only for average perturbed accuracy, but also for the attainment of formal robustness certificates.

# 6. Discussion

## 6.1. Efficiency of debiasing and certification

Based on Remark F.3, certification reduces to a one-dimensional problem, where if we have some small quantity of calibration perturbations or if perturbation statistics are known, even sampling from the uniform distribution becomes unnecessary. In addition, the proposed debiasing intervention is closed-form and efficient to compute. It integrates with the existing model structure, incurring negligible inference overhead.

## 6.2. Supervisory information

Debiasing for robustness can depend on the data and supervisory information available for the task at hand. Here we introduce both the unsupervised and supervised settings.

**Debias without supervision**: Input-dependent debiasing does not require supervision information in the form of class labels; what is needed instead is a clean example ( or a calibration set of clean examples). The unsupervised debiasing can improve performance, and as we show empirically, improves the rates of certification even for certification schemes requiring ground truth data to construct.

**LoRA as debiasing method**: LoRA can be used for debiasing with full supervisory information (ground truth labels for clean and perturbed examples). Figure 7 provides examples of LoRA and LoRA with input-dependent debiasing. Results indicate that LoRA is an effective debiasing method

if supervisory information is available. However, applying debiasing on top of LoRA may yield further gains in robustness and certification rates.

## 6.3. Tradeoff between clean and perturbed performance

Degradation of debiased model performance on clean data could be caused by excessive facilitation of "bad or irrelevant" features through debiasing, or through corrupting or even overwriting the impact of "relevant" attributes as a consequence of debiasing. Both causes could be controlled via a penalty term $\alpha \|\mathbf{\Psi}_c W\|^2$, where $\mathbf{\Psi}_c$ contains stacked $\psi(x) = \begin{bmatrix} x^\top, 1 \end{bmatrix}^\top$ and $x$ are features of clean (unperturbed) examples, added to the regression cost function:

$$\min_W \|\mathbf{\Psi} W - \mathbf{F}\|^2 + \lambda \|W\|^2 + \alpha \|\mathbf{\Psi}_c W\|^2 .$$

The corresponding solution, provided that the matrix $\mathbf{H} = \left( \mathbf{\Psi}^\top \mathbf{\Psi} + \alpha \mathbf{\Psi}_c^\top \mathbf{\Psi}_c + \lambda I_{n+2} \right)$ is non-singular, is:

$$W = \mathbf{H}^{-1} \mathbf{\Psi}^\top \mathbf{F}.$$

Introduction of the extra penalty term allows balancing between reducing projection of the weights on clean features (for $\alpha > 0$ ) and facilitating exploitation of "clean" features (for $\alpha < 0$) when other features' impact on clean performance is damaging. Parameter $\alpha$, therefore, enables control over the clean-perturbed trade-off, as shown in the following example in Figure 6 and Table 16 of the Appendix.

## 6.4. Generalization to generation tasks

For generalization to generation tasks, we move away from logits debiasing to debiasing features in intermediate layers. The approach is to identify the directions alongside which perturbations most consistently shift the last-token representation, and to debias projections to these directions. In application, this means minimizing expected clean-perturbed feature differences, performing PCA and debasing the top-$k$ principal components. If $k = 1$, this reduces to input-dependent debasing in Section 5.1.2. Preliminary evidence

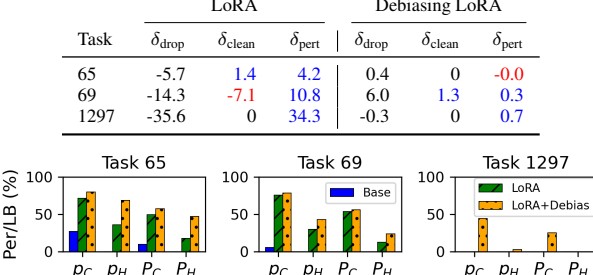

| Task | LoRA | | | Debiasing LoRA | | |
|---|---|---|---|---|---|---|
| | $\delta_{\text{drop}}$ | $\delta_{\text{clean}}$ | $\delta_{\text{pert}}$ | $\delta_{\text{drop}}$ | $\delta_{\text{clean}}$ | $\delta_{\text{pert}}$ |
| 65 | -5.7 | 1.4 | 4.2 | 0.4 | 0 | -0.0 |
| 69 | -14.3 | -7.1 | 10.8 | 6.0 | 1.3 | 0.3 |
| 1297 | -35.6 | 0 | 34.3 | -0.3 | 0 | 0.7 |

*Figure 7.* Impact of LoRA and debiasing on certification for tasks with format perturbations on Qwen 8B. LoRA adapters trained for layer 8 `mlp.up_proj` with clean + perturbed examples. Input-dependent debiasing applied on top of adapter-integrated model.

of this debiasing approach for generation and summarisation tasks are shown in Table 4 with details in Appendix G.4.

### 6.5. Positioning

We position our debiasing approach along three axis. First, in inference-cost, randomized smoothing (Cohen et al., 2019), self-denoising (Zhang et al., 2023) and template ensembling (Zhang et al., 2023) require multiple inference passes per single prompt. Our method adds no overhead beyond a single pass. Second, in application, these methods change input prompts, while ours act on output logits or intermediate features. Third, in perturbation model and certificates, the three target worst-case $l_2$-radius to guard against distribution-agnostic perturbations, whereas we address typical performance on non-adversarial nominal perturbations with data-aware certificates. Our method is closer in spirit to batch calibration (Zhou et al., 2023), but yields principled certificates and generalizes beyond classification.

### 6.6. Limitations

Perturbations examined in this work are not adversarial; they are semantically-neutral and randomly injected into prompts. Yet, their impact on performance was substantial. These results appear to be at odds with the literature on randomly perturbed models (cf. Sutton et al. 2024), where a radically different typical behavior of deep learning models was observed and explained. Further analysis revealed that this difference can be explained by the fact that effective intrinsic dimensions of feature spaces in each task are low, spanning 4-5 principal components. Whereas the robustness assurances in Sutton et al. (2024) assume high intrinsic dimensionality of data. This suggests that our method could be most efficient in relatively low-dimensional settings; as in genuinely high dimensions, concentration effects may mitigate the impact of systematic shifts.

Further limitations include: (1) empirical results focus largely on first-token prediction tasks enabling stringent vali-

*Table 4.* Example of debiasing in the intermediate layers for generalization to short-form generation tasks. Examples based on text perturbations with Qwen-8B. NQ refers to the Natural Questions benchmark, while EM refers to Exact Match metric.

| | NQ | GSM-8k | TriviaQA | XSum |
|---|---|---|---|---|
| $\delta_{\text{drop}}$ | 9.5 EM | 34.3 EM | 35.9 EM | 8.9 F1 |
| Layer 0 $p_{\text{recover}}$ | 104.5 | 63.3 | 87.5 | 123.1 |
| Layer 8 $p_{\text{recover}}$ | 105.5 | 68.8 | 83.4 | 111.8 |
| Layer 16 $p_{\text{recover}}$ | 79.6 | 59.4 | 58.4 | 56.0 |
| Layer 32 $p_{\text{recover}}$ | -31.4 | -79.5 | -33.7 | -125.9 |

dation of the theoretical analysis, but evaluation of debiasing in the intermediate layers and generalization to generation tasks was not as exhaustive; (2) the debiasing framework will result in a clean-perturbed input trade-off; more sophisticated debiasing methods might be designed to push the Pareto frontier on this trade-off; (3) certificates rely on distribution-agnostic concentration inequalities and assumptions that are conservative; (4) white-box access to model weights is required for debiasing.

## 7. Conclusion

Our work presents a principled framework for addressing the challenge of robustness of LLMs to semantically-neutral random prompt perturbations. Our theoretical analysis identifies a critical, yet previously underappreciated, factor of *perturbation-induced bias* – a systematic shift in expected module outputs under random perturbations.

Motivated by this insight, we propose debiasing approaches that compensate for these systematic shifts, are computationally efficient, do not require model retraining, and can operate without access to supervisory information. As we demonstrate in experiments, they enable recovering substantial portions of performance lost to perturbations.

Moreover, the proposed debiasing approaches stemming from our theoretical analysis strengthen formal robustness certificates that measure distances to decision boundaries: our interventions increase the proportion of per-example certifiable examples, and provide measurably stronger population-level robustness guarantees.

These results position our principled debiasing approaches as a practical intervention for enhancing LLM robustness in deployment settings where prompt variability is expected. Future work includes extending debiasing to long-form generation tasks, extending to application domains beyond languages, and developing tighter certificates.

## Impact Statement

This paper presents work aimed at improving the robustness and reliability of LLMs against random non-adversarial perturbations which are always expected at deployment but

which could be unaccounted for at the stage of training. The latter may occur due to the lack of computational resources or information about model deployment scenarios. One important feature of our method is that it can be applied in settings when ground truth information or data are not available at the model's training and robustness certification phases.

We believe this research contributes positively to making LLM deployments more trustworthy in real-world settings where inputs may contain formatting variations, typographical errors, or other semantically-neutral textual alterations. We empirically confirm that stock models from common architectural families may suffer from significant performance degradation induced by such random perturbations of prompts. Therefore, our theory, methods, and demonstration that interventions like debiasing can improve formal certification without requiring expensive full-model retraining, is practically impactful as it may improve trust in open-weight models in deployments to specific tasks.

## Acknowledgment

The work was supported by the RSF project 25-71-30008.

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

# A. Simple setting

In this section we provide theoretical analysis underpinning the development of the method.

The main element of the LLM's input-output mapping which will be the subject of our theoretical analysis is a $\gamma$-Lipschitz mapping $\mathbb{R}^n \to \mathbb{R}$. This mapping can represent input-output properties of any LLM module (MLP, Self-Attention), multiple modules stacked together, or even finer-grained relations between latent variables contained within each LLM module.

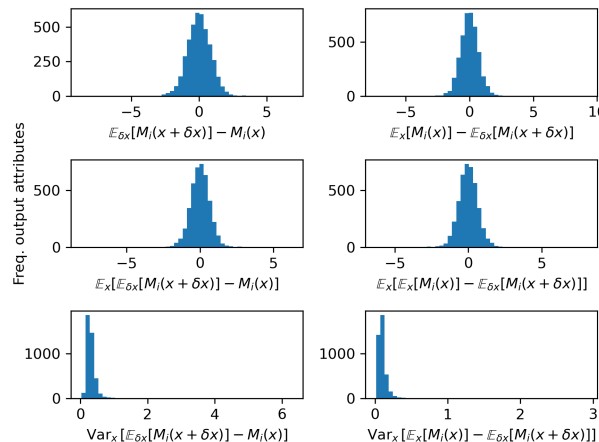

*Figure 8.* Measurements on perturbed formats of `Task 1284` of the `Natural-Instructions` dataset and Llama 8B model to validate geometric intuition in Figure 1. Here $M_i(x)$ corresponds to a single output attribute of the entire LLM before the logit layer, and the histograms show the distribution of measurements across all $n = 4096$ output attributes.

## A.1. Partial module setting

Assume the availability of a multi-set $S$ where

$$S = \{x_1, \ldots, x_m\}, \quad x_i \in \mathbb{R}^n, \quad i = 1, \ldots, m$$

Consider a module $M \in \mathcal{M}$ where $M : \mathbb{R}^n \to \mathbb{R}^d$ and $\mathcal{M}$ is a space of all potential modules. Each sub-module $M_i : \mathbb{R}^n \to \mathbb{R}$ for $i = 1, ..., d$ is $\gamma_i$-Lipschitz,

$$|M_i(x) - M_i(y)| \leq \gamma_i \|x - y\|$$

and provides the mapping to a single output attribute.

**Assumption A.1.** Perturbed inputs $\hat{x} \in \mathbb{R}^n$ are

$$\hat{x} = x + \delta x$$

where $\delta x = (\delta x_1, \ldots, \delta x_n)$ is the vector of independent random variables satisfying

$$\delta x_j \in [-c, c]$$

for some known $c > 0, c \in \mathbb{R}$

**Assumption A.2.** Let for each partial module, function $f_i : \mathbb{R}^n \times \mathbb{R}^n \to \mathbb{R}$ measure the difference between output attributes of base and perturbed inputs, i.e. for the $i$-th partial module,

$$f_i(x, \delta x) = M_i(x + \delta x) - M_i(x), \qquad (4)$$

and for each partial module, there exist and are known upper bounds $C_i$:

$$\sup_{x \in S} |\mathbb{E}_{\delta x}[f_i(x, \delta x)]| \leq C_i, \ i = 1, \ldots, d.$$

# B. Robustness certificates for the simple setting

**Theorem B.1** (Robustness certificate 1). *Consider the partial module setting with inputs $x \in S$ and perturbations $\delta x$ that satisfy Assumption A.1. Let functions $f_i(x, \delta x) = M_i(x + \delta x) - M_i(x)$ satisfy Assumption A.2.*

*Let $\mathcal{O}_i : \mathcal{M} \times \mathbb{R}^n \times \mathbb{R}^n \times \mathbb{R} \to \{0, 1\}$ be a testing map:*

$$
\mathcal{O}_i(M_i, x, \delta x, \epsilon_i) = \begin{cases} 1 & |f_i(x, \delta x)| \le \epsilon_i \\ 0 & \text{otherwise.} \end{cases}
$$

*Then for the partial module, if $\epsilon_i \ge C_i$, with probability over $\delta x$,*

$$
P(\mathcal{O}_i(M_i, x, \delta x, \epsilon_i) = 1 \text{ for all } x \in S)
$$
$$
\ge 1 - 2m \exp\left( -\frac{(\epsilon_i - C_i)^2}{2n(c\gamma_i)^2} \right).
$$

*Proof.* Based on the setting, for the partial module $i$, the function $f_i(x, \delta x)$ have the following bounded difference property,

$$
\sup_{\delta x_j \in \delta x} |f_i(x, \delta x_1, \ldots, \delta x_j, \ldots, \delta x_n)
$$
$$
- f_i(x, \delta x_1, \ldots, \delta x_j', \ldots, \delta x_n)| \le 2c\gamma_i
$$

since $M_i$ is $\gamma_i$-Lipschitz. McDiarmid's inequality implies that the following bound holds true:

$$
P(| f_i(x, \delta x) - \mathbb{E}[f_i(x, \delta x)] | \ge \xi)
$$
$$
\le 2 \exp\left( -\frac{\xi^2}{2n(c\gamma_i)^2} \right). \quad (5)
$$

Let us consider the event

$$
A : | f_i(x, \delta x) - \mathbb{E}[f_i(x, \delta x)] | \ge \xi.
$$

According to Assumption A.2,

$$
|\mathbb{E}_{\delta x}[f_i(x, \delta x)]| \le C_i.
$$

Hence if the event "not $A$" occurs, that is if $| f_i(x, \delta x) - \mathbb{E}[f_i(x, \delta x)] | < \xi$, then via triangular inequality,

$$
|f_i(x, \delta x)| \le |f_i(x, \delta x) - \mathbb{E}[f_i(x, \delta x)]|
$$
$$
+ |\mathbb{E}[f_i(x, \delta x)]| < \xi + C_i.
$$

Let $C$ be the event

$$
|f_i(x, \delta x)| \le \xi + C_i.
$$

The event "not $A$" implies that $C$ occurs. Therefore $P(\text{not } A) \le P(C)$. In other words,

$$
P(|f_i(x, \delta x)| < \xi + C_i) \ge
$$
$$
P(| f_i(x, \delta x) - \mathbb{E}[f_i(x, \delta x)] | < \xi). \quad (6)
$$

According to (5)

$$
P(|f_i(x, \delta x) - \mathbb{E}[f_i(x, \delta x)]| < \xi)
$$
$$
\ge 1 - 2\exp\left( -\frac{\xi^2}{2n(c\gamma_i)^2} \right). \quad (7)
$$

Letting $\epsilon_i = \xi + C_i$, recalling that $\epsilon_i \ge C_i$, and combining (6) with (7) results in:

$$
P(|f_i(x, \delta x)| < \epsilon_i) \ge 1 - 2\exp\left( -\frac{(\epsilon_i - C_i)^2}{2n(c\gamma_i)^2} \right).
$$

Given that the cardinality of the multi-set $S$ is $m$, the application of the union bound yields:

$$
P(|f_i(x, \delta x)| < \epsilon_i, \text{ for all } x \in S)
$$
$$
\ge 1 - 2m \exp\left( -\frac{(\epsilon_i - C_i)^2}{2n(c\gamma_i)^2} \right).
$$

This completes the proof. $\square$

For a linear partial module, a simple corollary is presented below in Appendix B.1.

*Remark* B.2 (Perturbation-induced bias, Lipschitz). $C_i$ captures the systematic bias in the output shift; it is the lower bound on the magnitude of the certificate. For fixed $\epsilon_i$, decreasing $C_i$ increases the confidence exponentially. For fixed target probability $P(\mathcal{O}_i = 1, \forall x \in S) = 1 - \phi_i$ where $\phi_i \in (0, 1]$ we have,

$$
\epsilon_i(\phi_i) = C_i + c\gamma_i \sqrt{2n \ln\left( \frac{2m}{\phi_i} \right)}. \quad (8)
$$

Reducing $C_i$ leads to a linear decrease in $\epsilon_i(\phi_i)$. Therefore, controlling $C_i$ leads to a linear control of the certified radius and an exponential gain in probability of certification for a fixed radius. It is therefore a vital factor for improving certified robustness.

*Remark* B.3. Eqn. 8 also captures the relationship between robustness verification multi-set $S$'s size $m$ and the Lipschitz constant $\gamma_i$. The robustness radius $\epsilon_i$ will scale with the size $m$ of the multi-set, but increase slowly with the square root of the natural logarithm. This can be balanced by adjusting the Lipschitz constant $\gamma_i$ (e.g., via direct scaling). On the other hand, $\epsilon_i$ scales directly with the Lipschitz constant $\gamma_i$ and therefore is essential to control for improving certified robustness.

We can also utilize the classical Chebyshev's inequality, which reveals another crucial parameter.

**Assumption B.4.** Let for each partial module, function $f_i : \mathbb{R}^n \times \mathbb{R}^n \to \mathbb{R}$ measure the difference between output attributes of base and perturbed inputs, i.e. for the $i$-th partial module,

$$
f_i(x, \delta x) = \mathcal{M}_i(x + \delta x) - \mathcal{M}_i(x).
$$

Moreover, for each partial module, there exist and are known upper bounds $C_i, V_i$:

$$\sup_{x \in S} |\mathbb{E}_{\delta x} [f_i(x, \delta x)]| \leq C_i, \ i = 1, \ldots, d,$$

$$\sup_{x \in S} |\mathrm{Var}_{\delta x} [f_i(x, \delta x)]| \leq V_i, \ i = 1, \ldots, d.$$

**Theorem B.5** (Robustness certificate 2)**.** *Consider any partial module setting with inputs $x \in S$. Let functions $f_i(x, \delta x) = M_i(x + \delta x) - M_i(x)$ satisfy Assumption B.4.*

*Let $\mathcal{O}_i : \mathcal{M} \times \mathbb{R}^n \times \mathbb{R}^n \times \mathbb{R} \to \{0, 1\}$ be a testing map:*

$$\mathcal{O}_i(M_i, x, \delta x, \epsilon_i) = \begin{cases} 1 & |f_i(x, \delta x)| \leq \epsilon_i \\ 0 & otherwise. \end{cases}$$

*Then for the partial module, if $\epsilon_i \geq C_i$, with probability over $\delta x$,*

$$P(\mathcal{O}_i(M_i, x, \delta x, \epsilon_i) = 1 \text{ for all } x \in S)$$
$$\geq 1 - m \frac{V_i}{(\epsilon_i - C_i)^2}.$$

*Proof.* Based on Chebyshev inequality,

$$P\left(|f_i(x, \delta x) - \mathbb{E}[f_i(x, \delta x)]| \geq t\right)$$
$$\leq \frac{\mathrm{Var}[f_i(x, \delta x)]}{t^2}$$

for any $t > 0$. Using triangular inequality like in the proof of Theorem B.1, we can let $\epsilon_i = t + C_i$, and with $\epsilon_i \geq C_i$

$$P\left(|f_i(x, \delta x)| \geq \epsilon_i\right) \leq \frac{\mathrm{Var}[f_i(x, \delta x)]}{(\epsilon_i - C_i)^2}.$$

Given the cardinality of the multi-set $S$ is $m$ and $\mathrm{Var}[f_i(x, \delta x)] \leq V_i$ for $x \in S$, the application of the union bound yields,

$$P\left(|f_i(x, \delta x)| < \epsilon_i, \text{ for all } x \in S\right)$$
$$\geq 1 - m \frac{V_i}{(\epsilon_i - C_i)^2}.$$

This completes the proof. $\square$

*Remark* B.6 (Perturbation-induced bias, variance). Theorem B.5 places no direct assumption on the magnitude of noise $\delta x$ or the Lipschitz constant $\gamma_i$ of the partial module. Intuitively, it suggests that by controlling both the supremum related to expectation $C_i$ and variance $V_i$ of change (across perturbations), we improve the probability of certification for fixed $\epsilon_i$. For a fixed target bound on the probability $P(\mathcal{O}_i = 1, \forall x \in S)$:

$$P(\mathcal{O}_i = 1, \forall x \in S) \geq 1 - \phi_i$$

where $\phi_i \in (0, 1]$, we have:

$$\epsilon_i(\phi_i) = C_i + \sqrt{\frac{V_i m}{\phi_i}}. \tag{9}$$

Decreasing $C_i$ results in a linear decrease in $\epsilon_i(\phi_i)$, and decreasing $V_i$ reduces the $\epsilon_i(\phi_i)$ proportional to the square root of $V_i$. It is also important to note that the required tolerance grows with the square root of the size $m$ of the multi-set $S$, which will become a significant limitation for a large set $S$ unless $V_i$ is controlled.

### B.1. Linear partial module

**Corollary B.7.** *Consider the partial module with inputs $x \in S$ and perturbations $\delta x$ that satisfy Assumption A.1. If the $i$-th partial module is linear,*

$$M_i(x) = W_i x + b, \ W_i \in \mathbb{R}^{1 \times n}, b \in \mathbb{R}$$

*If an oracle $\mathcal{O}_i : \mathcal{M} \times \mathbb{R}^n \times \mathbb{R}^n \to \{0, 1\}$ is available for each partial module, then for the partial module, if $\epsilon_i \geq C_i$, with probability over $\delta x$,*

$$P(\mathcal{O}_i(M_i(\cdot), x, \delta x) = 1 \text{ for all } x \in S)$$
$$\geq 1 - 2m \exp\left(-\frac{(\epsilon_i - C_i)^2}{2n(c\|W_i\|)^2}\right)$$

*Proof.* The proof follows from proof of Theorem B.1, but since the partial module is linear, $\gamma_i = \|W_i\|$. $\square$

## C. Bias correction

For control of $C_i$, a simple approach is to add a bias $b_i \in \mathbb{R}$ to the partial module $M_i(x)$, which gives us a corrected module,

$$M_i^*(x) = M_i(x) - b_i.$$

Define function $f_i^* : \mathbb{R}^n \to \mathbb{R}$ that measures the difference between output attributes of base inputs from the original partial module and perturbed inputs from the corrected module, i.e. for the $i$-th corrected partial module,

$$f_i^*(x, \delta x) = M_i^*(x + \delta x) - M_i(x). \tag{10}$$

**Assumption C.1.** For each corrected partial module, there exists known upper bounds $C_i^*(b_i) \geq 0$:

$$\sup_{x \in S} |\mathbb{E}_{\delta x} [f_i(x, \delta x)] - b_i| \leq C_i^*(b_i).$$

**Theorem C.2** (Corrected partial module)**.** *Consider the partial modules $M_i \in \mathcal{M}, M_i^* \in \mathcal{M}^*$ which satisfy Assumption C.1, with inputs $x \in S$ and perturbations $\delta x$ that satisfy Assumption A.1.*

Let $\mathcal{O}_i^* : \mathcal{M} \times \mathcal{M}^* \times \mathbb{R}^n \times \mathbb{R}^n \times \mathbb{R} \to \{0, 1\}$ be a testing map:

$$\mathcal{O}_i^*(M_i, M^*, x, \delta x, \epsilon_i) = \begin{cases} 1 & |f_i^*(\delta x)| \leq \epsilon_i \\ 0 & otherwise. \end{cases}$$

Then for $\epsilon_i \geq C_i^*(b_i)$

$$P(\mathcal{O}_i^*(M_i, M_i^*, x, \delta x, \epsilon_i) = 1, \forall x \in S)$$
$$\geq 1 - 2m \exp\left(-\frac{(\epsilon_i - C_i^*(b_i))^2}{2n(c\gamma_i)^2}\right)$$

*Proof.* The proof follows immediately from the argument presented in the proof of Theorem B.1. □

*Remark* C.3. For bias correction, the bias can also be input-dependent,
$$M_i^*(x) = M_i(x) - b_i(x)$$

The rationale is that by optimizing the value of

$$\sup_{x \in S} |\mathbb{E}_{\delta x} f_i(x, \delta x) - b_i(x + \delta x)|,$$

over $b_i \in \mathcal{J}$, where $\mathcal{J}$ is a class of functions $\mathbb{R}^n \to \mathbb{R}$ containing all constant functions, one can potentially achieve a more efficient compensation of $C_i$ and $V_i$. Indeed

$$\inf_{b_i \in \mathcal{J}} \sup_{x \in S} |\mathbb{E}_{\delta x} f_i(x, \delta x) - b_i(x + \delta x)|$$
$$\leq \inf_{\theta_i \in \mathbb{R}} \sup_{x \in S} |\mathbb{E}_{\delta x} f_i(x, \delta x) - \theta_i|$$
$$\leq \sup_{x \in S} |\mathbb{E}_{\delta x} f_i(x, \delta x)|.$$

This provides significantly more flexible bias correction.

# D. General Setting

In the prior sections on the simple setting, we worked on partial modules $M_i$ with any product perturbation distribution with bounded support and coordinate-wise independent perturbations. From this section onward, we generalise to perturbation distribution $D$ that can exhibit dependencies.

In order to develop certificates accounting for model performance in a given task, we focus on the general setting with class or token margins.

## D.1. Data model

Consider $(x, y) \sim \mathcal{X} \times \mathcal{Y}$ with data $x \in \mathbb{R}^n$ and label $y \in \{1, \ldots, Q\}$ for a classification problem with $Q$ classes.

Data can be inputs into the model, be immediate outputs of the LLM's tokenizer, outputs of model's intermediate layers, or outputs from the penultimate layer of the model.

## D.2. Model of perturbations

Perturbations to input data $(x, y)$ are assumed to be random variables $\delta x$, whose components are dependent. More formally, for an original data vector $x$, a perturbed data is

$$\hat{x} = x + \delta x,$$

where $\delta x$ is sampled from any distribution $D$ on $\mathbb{R}^n$, including the ones that may be dependent on $x$ and $y$.

Fix a box in $\mathbb{R}^n$,

$$\mathcal{B} := \prod_{i=1}^n [l_i, u_i]$$

with $u_i - l_i > 0$, and volume $V_\mathcal{B} := \prod_{i=1}^n (u_i - l_i)$.

**Assumption D.1.** For fixed $x$ and box $\mathcal{B}$, there exist constants $C \geq 0$, $\epsilon_\mathcal{B} \in (0, 1)$ such that for any $A \subset \mathbb{R}^n$

$$P_D(A \cap \mathcal{B}) \leq CV_\mathcal{B} P_u(A \cap \mathcal{B}),$$

where $P_u$ is the uniform product distribution on $\mathcal{B}$, and

$$P_D(A \cap \mathcal{B}^c) \leq \epsilon_\mathcal{B}.$$

Assumption D.1, in essence, requires that the distribution $P_D$, if supported on a bounded domain, does not have pathological concentrations in small volumes over sufficiently large domains in its support represented by the box $\mathcal{B}$. If the support of $P_D$ is not bounded, then this assumption requires that the "bulk" of the measure is inside $\mathcal{B}$. This is controlled by the value of $\epsilon_\mathcal{B}$.

**Lemma D.2.** *Fix $x$ and a box $\mathcal{B}$, and suppose that Assumption D.1 holds with constants $C$ and $\epsilon_\mathcal{B}$. Then for any $A \subset \mathbb{R}^n$ the following bound holds true:*

$$P_D(A) \leq CV_\mathcal{B} P_u(A) + \epsilon_\mathcal{B}$$

*where $P_u$ is the uniform product distribution on $\mathcal{B}$*

*Proof.* Since from Assumption D.1 we know that $P_D(A \cap \mathcal{B}) \leq CV_\mathcal{B} P_u(A \cap \mathcal{B})$ and $P_D(A \cap \mathcal{B}^c) \leq \epsilon_\mathcal{B}$, by the law of total probability,

$$P_D(A) = P_D(A \cap \mathcal{B}) + P_D(A \cap \mathcal{B}^c)$$
$$\leq CV_\mathcal{B} P_u(A \cap \mathcal{B}) + \epsilon_\mathcal{B}$$

where $P_u(A \cap \mathcal{B}) = P_u(A)$ since $P_u$ is supported on $\mathcal{B}$, which completes the proof. □

## D.3. Margins

Consider linear mappings $M_i : \mathbb{R}^n \to \mathbb{R}$ for classes $i = 1, \ldots, Q$:
$$M_i(x) = \langle w_i, x \rangle + b_i.$$
The geometric margin associated with $M_i$ and a label $y \in \mathcal{Y}$, $y \neq i$ in Euclidean space is:

$$\rho(x, y) = \min_{i \neq y} \frac{\langle w_y - w_i, x \rangle + (b_y - b_i)}{\|w_y - w_i\|}.$$

## D.4. Margin shift due to noise

Define the shift in margin due to noise as,

$$f(x, y, \delta x) = \rho(x + \delta x, y) - \rho(x, y)$$

**Assumption D.3.** For the given $(x, y)$ and $\mathcal{B}$, the shift in margin lies within a fixed effective range,

$$f(x, y, \delta x) \in [-c(x, y), c(x, y)] \tag{11}$$

where

$$c(x, y) := \sup_{\delta x \in \mathcal{B}} |f(x, y, \delta x)|.$$

## D.5. Per-example statistics

Define for a fixed $(x, y)$, under uniform distribution $u$

$$\mu_u(x, y) := \mathbb{E}_u \left[ f(x, y, \delta x) \right]$$
$$\nu_u(x, y) := \text{Var}_u \left[ f(x, y, \delta x) \right]$$

Condition for correct classification is $\rho(x, y) > 0$ for an unperturbed example and $\rho(x + \delta x, y) = \rho(x, y) + f(x, y, \delta x) > 0$ for a perturbed example.

# E. Generalization of results from the simple setting

Here we show how results presented in Appendix A-C for the simple settings (bounded support and coordinate-wise independent perturbations) can be generalized to the general setting, where the true perturbation distribution $D$ can exhibit dependencies and can also depend on $(x, y)$.

Assumption D.1 and Lemma D.2 provide the tools for this generalization by introducing a box $\mathcal{B}$ and a product reference distribution $P_u$ which can be the uniform product distribution on $\mathcal{B}$. According to Lemma D.2, for any event $A$, the relationship between the true and unknown distribution $P_D$ and the reference product distribution $P_u$ can be expressed as:

$$P_D(A) \leq CV_{\mathcal{B}} P_u(A) + \epsilon_{\mathcal{B}}.$$

Then any certificate from the simple setting bounding the probability of a failure event under some reference distribution $P_u$ can be immediately transferred to the true distribution. In particular, for any target failure ratio $\phi \in (0, 1)$ under true perturbation distribution $D$, it is sufficient to request that the following bound holds true:

$$P_u(A) \leq \frac{\phi - \epsilon_{\mathcal{B}}}{CV_{\mathcal{B}}}.$$

To illustrate this possibility, we present an example of a possible generalization from the simple-setting robustness certificate presented in the earlier Section B.

Consider Theorem B.1. Let $A$ be the event that

$$\mathcal{O}_i(M_i, x, \delta x, \epsilon_i) = 0, \text{for some } x \in S$$

According to this theorem,

$$P_u(A) \leq 2 \exp \left( -\frac{(\epsilon_i - C_i)^2}{2n(c\gamma_i)^2} \right)$$

where the box is defined as the hypercube $\mathcal{B} = [-c, c]^n$. By Assumption D.1 and Lemma D.2,

$$P_D(A) \leq 2CV_{\mathcal{B}} \exp \left( -\frac{(\epsilon_i - C_i)^2}{2n(c\gamma_i)^2} \right) + \epsilon_{\mathcal{B}}$$

Hence

$$P(\mathcal{O}_i(M_i, x, \delta x, \epsilon_i) = 1 \text{ for all } x \in S)$$
$$\geq 1 - 2CV_{\mathcal{B}} m \exp \left( -\frac{(\epsilon_i - C_i)^2}{2n(c\gamma_i)^2} \right) - m\epsilon_{\mathcal{B}}.$$

Other statements from the earlier Sections A-C can be generalized in the same manner.

In the next section we present further results enabling explicit certification of robustness for classification and token prediction tasks in terms of margins.

# F. Certificates for margins in the general setting

**Theorem F.1.** *Suppose Assumptions D.1 and D.3 hold true with parameters $\epsilon_{\mathcal{B}} \in (0, 1)$, $c(x, y)$, $C$ and $V_{\mathcal{B}}$ for a fixed example $(x, y)$. Furthermore, let*

$$\rho(x, y) > -\mu_u(x, y) + \min \{\xi_H(x, y), \xi_B(x, y)\},$$

*where*

$$\xi_H(x, y) = c(x, y)\sqrt{2\kappa}$$
$$\xi_B(x, y) = \gamma(x, y) + \sqrt{\gamma(x, y)^2 + 2\kappa\nu_u(x, y)}$$

*and*

$$\kappa := \ln \left( \frac{CV_{\mathcal{B}}}{\phi - \epsilon_{\mathcal{B}}} \right), \quad \gamma := \frac{2}{3} c(x, y)\kappa.$$

*Finally, let $\phi \in (0, 1)$ and $\phi > \epsilon_{\mathcal{B}}$.*

*Then the classification on the noisy example is correct with probability at least $1 - \phi$,*

$$P_D(\rho(x + \delta x, y) > 0) \geq 1 - \phi. \tag{12}$$

*Proof.* For any fixed $(x, y)$, define the misclassification event,

$$E := \{\delta x \in \mathbb{R}^n : \rho(x + \delta x, y) \leq 0\}.$$

then
$$P_D(\rho(x + \delta x, y) > 0) = 1 - P_D(E)$$

so we want to prove

$$P_D(E) \leq \phi. \qquad (13)$$

Let constant $C \geq 0$ and $\mathcal{B}$ satisfy Assumption D.1. Then by Lemma D.2 the probability of the misclassifications event $E$ is bounded,

$$P_D(E) \leq C V_{\mathcal{B}} P_u(E) + \epsilon_{\mathcal{B}}.$$

Pick a $\phi > \epsilon_{\mathcal{B}}$ and

$$\kappa := \ln\left(\frac{C V_{\mathcal{B}}}{\phi - \epsilon_{\mathcal{B}}}\right).$$

Then if we can show that

$$P_u(E) \leq e^{-\kappa}, \qquad (14)$$

we consequently prove

$$P_D(E) \leq C V_{\mathcal{B}} e^{-\kappa} + \epsilon_{\mathcal{B}} = \phi$$

which implies

$$P_D(\rho(x + \delta x, y) > 0) > 1 - \phi$$

So the proof reduces to the proof of Eqn. 14.

To prove Eqn. 14, define random variable

$$X := f(x, y, \delta x) = \rho(x + \delta x, y) - \rho(x, y)$$

then
$$\begin{aligned} P_u(E) &= P_u(X \leq -\rho(x, y)) \\ &= P_u(X - \mu_u(x, y) \leq -t) \end{aligned}$$

with $t := \rho(x, y) + \mu_u(x, y)$, and $\mu_u(x, y) = \mathbb{E}_u[X]$, $\nu_u(x, y) = \text{Var}[X]$.

According to Assumption D.3,

$$X \in [-c(x, y), c(x, y)]$$

for all $\delta x \in \mathcal{B}$. We can therefore use the Hoeffding's inequality for bounded random variable $X \in [-c, c]$,

$$P_u(E) = P_u(X - \mu_u(x, y) \leq -t) \leq e^{-\frac{t^2}{2c^2}}.$$

Hence, in order to achieve $P_u(E) \leq e^{-\kappa}$, it is sufficient to pick
$$t \geq c(x, y)\sqrt{2\kappa}.$$

Therefore, setting $t = \rho(x, y) + \mu_u(x, y)$, or alternatively, requesting that

$$\rho(x, y) \geq -\mu_u(x, y) + \xi_H(x, y) \qquad (15)$$

where

$$\xi_H(x, y) = c(x, y)\sqrt{2\kappa},$$

demonstrates that $P_u(E) \leq e^{-\kappa}$.

Instead of Hoeffding inequality one can employ Bernstein inequality for bounded random variables. Applying the same logic results in

$$\rho(x, y) \geq -\mu_u(x, y) + \xi_B(x, y) \qquad (16)$$

with

$$\xi_B(x, y) = \gamma(x, y) + \sqrt{\gamma(x, y)^2 + 2\kappa \nu_u(x, y)}$$

and $\gamma := \frac{2}{3} c(x, y)\kappa$.

Satisfying either ( 15) or ( 16) implies ( 14) and consequently ( 12). Hence, taking the minimum,

$$\xi(x, y) = \min\{\xi_H(x, y), \xi_B(x, y)\}$$

completes the proof.

$\square$

*Remark* F.2. Theorem F.1 can be easily adapted to utilize other concentration inequalities, including the Chebyshev or Cantelli inequalities. These inequalities are more distribution dependent and less conservative, as shown empirically in certification percentage and guarantee lower bounds offered in Section 5.2.

For binary classification problems and any $\mathcal{B}$ centered at $c^* \in \mathbb{R}^n$ the value of $\mu_u(x, y)$ could be computed as:

$$\begin{aligned} \mathbb{E}_u(f(x + \delta x, y)) &= \frac{\langle w_y - w_{i \neq y}, x \rangle + (b_y - b_{i \neq y})}{\|w_y - w_{i \neq y}\|} \\ &+ \frac{\langle w_y - w_{i \neq y}, c^* \delta x \rangle}{\|w_y - w_{i \neq y}\|} = \mu_u(x, y). \end{aligned}$$

To provide sample-wise guarantee, noise sampling is required in inference. Assume for a single fixed example $(x, y)$, the availability of $z$ i.i.d. perturbations based on the uniform product distribution $u$ on $\mathcal{B}$ for calibration. This enables computing empirical mean over shifts in margins at a fixed $(x, y)$ induced by perturbations from the uniform product distribution:

$$\hat{\mu}_u(x, y) = \frac{1}{z} \sum_{j=1}^{z} f(x, y, \delta x^{(j)}).$$

*Remark* F.3 (Certificates for linear modules). Consider a linear classifier (with no bias) for classification with classes $i \in \{0, \cdots, Q\}$,

$$M_i(x) = \langle w_i, x \rangle$$

then we can redefine the pairwise margin for $j \neq y$:

$$\rho_j(x, y) = \langle u_{y,j}, x \rangle, \text{ with } u_{y,j} := \frac{w_y - w_j}{\|w_y - w_j\|}$$

and then for inputs with perturbations, the perturbations can be projected to 1D:

$$Z_j := \rho_j(x + \delta x, y) - \rho_j(x, y) = \langle u_{y,j}, \delta x \rangle$$

For each $j \neq y$ we can define a box $\mathcal{B}_j$ (an interval in 1D), where $\mathcal{B}_| = [l_j, u_j]$ with volume $V_\mathcal{B} = u_j - l_j$ and out-of-box probability of $\epsilon_{\mathcal{B}_j}$.

Under the uniform product distribution on box $\mathcal{B}_j$ we know explicit true mean $\mu_{u_j} = V_\mathcal{B}/2$ and variance $\nu_{u_j} = V_\mathcal{B}^2/12$ and the boundness constant $c_j = (u_j - l_j)/2$ without the need of sampling. From these we can define

$$\kappa_j := \ln\left(\frac{C_j V_{\mathcal{B}_j}}{\phi_j - \epsilon_{\mathcal{B}_j}}\right), \quad \gamma_j := \frac{2}{3} c_j \kappa_j$$

and

$$\xi_j = \min\left\{c_j\sqrt{2\kappa_j}, \gamma_j + \sqrt{\gamma_j^2 + 2\kappa_j \nu_{u_j}}\right\}$$

Let Assumption D.1. and D.3 hold true with the parameters in the above setting. Then if for all $j \neq y$, let $\phi_j \in (0, 1)$ and $\phi_j > \epsilon_{\mathcal{B}_j}$ the following condition is satisfied:

$$\rho_j(x, y) \geq -\mu_{u_j} + \xi_j$$

then the probability that the classification is correct for example with perturbation is lower bounded with $\phi = \sum_{j \neq y} \phi_j$:

$$P_D(\rho(x + \delta x, y) > 0) \geq 1 - \phi.$$

The per-example guarantee will require per-example sampling of perturbations or information on the perturbation distribution for certification. We can generalize this to population guarantees for samples from a data distribution.

*Remark* F.4 (Population-level certificates). Suppose that the examples are sampled from a distribution $D'$ and the perturbation $\delta x \sim D$ is sampled independently of $(x, y) \sim D'$. Furthermore, assume the availability of a calibration set $S$ containing $m$ examples sampled i.i.d. from $D'$, where a fraction $s/m$ of the examples is certified in the per-example level. Then one can generalize example-level certificates to the population-level ones. Indeed, let $\psi \in (0, 1)$. Then with probability at least $1 - \psi$ over the draw of $S$ Hoeffding's inequality implies that the following bound holds:

$$P_{\delta x \sim D, (x,y) \sim D'}(\rho(x + \delta x, y) = 1)$$

$$\geq (1 - \phi)\max\left\{0, \frac{s}{m} - \sqrt{\frac{1}{2m}\ln\left(\frac{2}{\psi}\right)}\right\}.$$

# G. Supplementary details on experiments

## G.1. Tasks from the Natural Instructions dataset

List of individual task indices, names and descriptions from the Natural Instructions (Wang et al., 2022). These are the results of the task selection protocol of Seleznyov et al. (2025) with additional task filters: (1) a $> 1\%$ drop in performance due to perturbations, (2) balanced accuracy (BAC) $\geq 70\%$ on original examples from the benchmark (*clean* examples). The additional task filters filters for tasks for whcih the model is performant on clean examples and is noticeably less accurate under semantically-neutral prompt alternations.

- **Task 065 Time-travel consistent sentence classification -** Choosing the option that makes a given short story consistent

- **Task 069 Abductive NLI classification -** Choosing text that completes a story based on given beginning and ending.

- **Task 1297 QASC question answering -** Given two facts, and a multiple-choice question, answer the question.

- **Task 280 Stereo-set classifcation -** Classify sentences into four kinds of stereotypes, including gender, profession, race, and religion.

- **Task 286 Openbook-QA -** Given a multiple-choice question and you have to pick the correct option.

- **Task 296 Story-cloze correct end classification -** Given four sentences of five sentence story, select correct answer for last (fifth) sentence from the given option.

- **Task 220 ROC-stories title classification -** Given a five sentence story, choose an appropriate title for the story from the given options.

- **Task 158 Count frequency of words -** Count the number of occurrences of a word in the given sentence.

- **Task 322 Jigsaw classification threat -** Given a comment from online platforms, classify whether or not it contains threats.

## G.2. Data splits

For each target task, we sample 200 examples and generate 200 perturbations per example for each perturbation type. This is split into three subsets: 50% training set, 35% testing set and an additional 15% hold-out set. The main body of

---

**Original**

```
Sentence 1: Marion and Louise had a pet parrot named Preacher…
Sentence 5: I said hello back, and he asked how I was doing
Option 1: When I walked in, Preacher said hello.
Option 2: When I walked in, Preacher said hello but will never
    speak again.
```

---

**Example format perturbation 1**

```
Sentence 1) : Marion and Louise had a pet parrot named
    Preacher…
Sentence 5) : I said hello back, and he asked how I was doing ,
Option i ) : When I walked in, Preacher said hello.
Option ii ) : When I walked in, Preacher said hello but will
    never speak again. ,
```

---

**Example format perturbation 2**

```
Sentence [I]- Marion and Louise had a pet parrot named
    Preacher. --…
Sentence [V]- I said hello back, and he asked how I was doing
    --
Option i. When I walked in, Preacher said hello.
Option ii. When I walked in, Preacher said hello but will never
    speak again. --
```

---

**Example text perturbation 1**

```
Sentence 1: Marion and Louise had a pet parrot named Preacher…
Sentence 5: I\xa0said hello back, and he asked how I was doing
Option 1: When I walked in, Preacher said hello.
Option 2: When I walked in,\xa0Preacher said hello but will
    never speak again.
```

---

**Example text perturbation 2**

```
Sentence 1: Marion and Louise had a pet par \u200drot named
    Preacher…
Sentence 5: I\xa0said hello back, and he asked how I\xa0was
    doing
Option 1: When I walked in, Preacher sai \u200cd hello.
Option 2: When I walked in, Preacher sa \ufeffid hello but
    wi\u200cll never speak again.
```

---

*Figure 9.* Examples of semantically similar but textually different prompts for a random example from Task 065.

results presented are based on the measurements on the testing set, while the population-level certificates are measured using the hold-out set.

For adequate split w.r.t. perturbations:

- For text perturbations, since perturbations can occur in any location in the string, random text perturbations are applied independently for the examples of each set.

- For format perturbations, since perturbations are rigid and in fixed locations in the string, a fixed set of 200 formats was considered. These formats have been split randomly into the training set (50%), testing set (35%) and hold-out (15%), similar to the example-level split. For each set, only the formats in the corresponding

format split set were applied to ensure no data leakage occurs through the use of same formats in both training and testing.

For per-example certification, values of $l_i, u_i$ are estimated empirically with 50% of perturbed examples and evaluated on the remaining 50%. For multi-class tasks, each pair of ground-truth vs competitor class are certified separately; an example is certified only if all pairwise certificates pass.

### G.3. Ridge Regression to $f_i$

For example, we can define

$$b_i(x) = w_i \cdot \psi(x) + \beta_i$$

where $\psi(x) = \begin{bmatrix} x^\top, \|x\|_2, 1 \end{bmatrix}^\top \in \mathbb{R}^{(n+2)}$, $\beta_i \in \mathbb{R}$ is a constant, and $w_i \in \mathbb{R}^{(n+2) \times 1}$ (where $n$ is the input dimension $x \in \mathbb{R}^n$). Then we can define the inputs and targets of the ridge regression:

- Inputs: For $N$ training prompts and $K$ perturbed variants of each training prompt, we can stack feature mappings $\psi(x)$ into $\Psi \in \mathbb{R}^{NK \times (n+2)}$.

- Targets: For each feature mapping, we can find $f_i(x, \delta x) = M_i(x + \delta x) - M_i(x)$ for partial modules $i = 1, ..., d$, which can also be stacked to target outputs of $\mathbf{F} \in \mathbb{R}^{NK \times d}$

The regression problem becomes,

$$\min_W \|\boldsymbol{\Psi} W - \mathbf{F}\|^2 + \lambda \|W\|^2$$

for $W = [w_1, ..., w_d] \in \mathbb{R}^{(n+2) \times d}$, where $\lambda \in (0, 1]$ is the regularisation constant. We use the closed-form solution of

$$W = \left(\boldsymbol{\Psi}^\top \boldsymbol{\Psi} + \lambda I_{n+2}\right)^{-1} \boldsymbol{\Psi}^\top \mathbf{F}$$

which gives residuals:

$$R = \mathbf{F} - \boldsymbol{\Psi} W$$

used to define the additional constant:

$$\beta_i = \frac{1}{2} \left( \max_j R_{j,i} + \min_j R_{j_i} \right)$$

where $j$ is index over $NK$ single prompt residuals, and choice of $\beta_i$ is to reduce residuals:

$$\min_{\beta_i} \max_j |R_{j,i} - \beta_i|$$

In the empirical experiments, we solve the problem with Cholesky factorization with adaptive jitter.

*Remark* G.1 (Linear ridge). This previous form of ridge regression is non-linear, but can be easily generalized to the linear case with

$$\psi(x) = \begin{bmatrix} x^\top, 1 \end{bmatrix}^\top \in \mathbb{R}^{(n+1)}$$

which will allow merging of the weights into the original logit layer. In the empirical experiments, we utilize linear Linear regression.

### G.4. Generalisation to generation tasks

To extend from classification tasks to generation tasks, we can move away from debiasing the logits towards debiasing features in intermediate layers.

An example approach is to select an intermediate layer, extract $\mathbb{R}^n$ feature vectors of the last token before generation for both clean and perturbed input prompts, compute the differences between clean and perturbed feature vectors for the training set and their mean $\mu$, define $\phi(x) = x - \mu$, for feature vector $x$, (3) compute PCA on the training set comprising $\phi(x)$, (4) select $k$ principal components and perform debiasing on the projections to these $k$ principal components.

If $H = (h_1, \ldots, h_n)$ is the matrix of principal components, and $\theta : \mathbb{R}^n \to \mathbb{R}^k$ is the debiasing function (such as the ridge regression) then intermediate layer debiasing can be achieved as:

$$\phi_{\text{debiased}}(x) = H \begin{pmatrix} \theta_1 \left( H^T \phi(x) \right) + h_1^T \phi(x) \\ \vdots \\ \theta_k \left( H^T \phi(x) \right) + h_k^T \phi(x) \\ h_{k+1}^T \phi(x) \\ \vdots \\ h_n^T \phi(x) \end{pmatrix}$$

where for $k = 1$, this reduces the problem precisely to the one we analyzed for input-dependent debiasing.

An experiment of the above debiasing approach have been run for simple generation (NQ, TriviaQA, GSM-8K ) and summarisation (XSum) tasks. For each of these tasks we sampled 275 example subsets and generated 100 text perturbations for each example. The train-test split is 100/175 for examples, and 30/70 for perturbations. We set $k = 1$, so that debiasing was applied only to projections on the first principal component, leaving other variables untouched. The results is shown in Table 4 to demonstrate this potential generalisation.

## H. Supplementary empirical results

This section presents supplementary empirical results:

*Table 5.* The effect of input-independent debiasing on binary tasks with format perturbations (top section) and text perturbations (bottom section). Extended table of Table 1. Here red and blue indicate decrease and increase in performance/robustness due to debiasing, respectively. Negative $p_{\epsilon V}$ and $p_{\epsilon L}$ indicate expansion of robustness radii.

| Model | Task | $\delta_{\text{drop}}$ | $\delta_{\text{clean}}$ | $\delta_{\text{pert}}$ | $p_{\epsilon L}$ | $p_{\epsilon V}$ |
|---|---|---|---|---|---|---|
| Llama 8B | 065 | -20.9 | -10.1 | 11.1 | 75.9 | 53.1 |
| Llama 8B | 069 | -19.3 | -8.3 | 12.8 | 68.5 | 67.5 |
| Llama 8B | 286 | -4.7 | 0 | 0.5 | 58.6 | -62.9 |
| Llama 8B | 296 | -35.1 | – | – | 60.3 | -83.4 |
| Qwen 8B | 065 | -5.7 | – | – | 25.7 | -67.4 |
| Qwen 8B | 069 | -14.3 | – | – | -2.8 | -18.2 |
| Qwen 8B | 220 | -6.3 | -1.4 | -1.1 | 51.7 | 14.4 |
| Qwen 8B | 296 | -23.3 | – | – | 11.4 | -16.3 |
| Olmo 7B | 065 | -13.4 | – | – | 38 | 22.9 |
| Olmo 7B | 069 | -13.3 | – | – | 62.4 | -4.8 |
| Olmo 7B | 220 | -18.9 | – | – | 15.9 | -39.8 |
| Olmo 7B | 296 | -14.6 | – | – | 16.6 | -13.8 |
| Olmo 7B | 322 | -2.4 | – | – | 51.5 | -34.4 |
| Qwen 8B | 065 | -6.1 | – | – | 15.6 | -76.9 |
| Qwen 8B | 220 | -1.7 | -2.7 | 0.6 | 76.9 | -92.8 |
| Llama 8B | 065 | -16.7 | – | – | 73.8 | 61.1 |
| Llama 8B | 069 | -13.3 | 0.9 | 6.5 | 68.5 | 45.6 |
| Llama 8B | 286 | -14.8 | – | – | 14.5 | -43.1 |
| Llama 8B | 296 | -19.8 | – | – | 42.7 | -10.1 |
| Olmo 7B | 065 | -21.8 | 2.7 | 7.8 | 35.5 | -11.6 |
| Olmo 7B | 069 | -19.6 | – | – | 5 | -24.4 |
| Olmo 7B | 1284 | -19.9 | – | – | 59.3 | 32.3 |
| Olmo 7B | 322 | -14.8 | – | – | 30.8 | 1.5 |

- Table 5-9 provides supplementary empirical results for input-independent and input-dependent debiasing.

- Table 10-15 provides supplementary empirical results for per-example and population level certification, including the effect of input-dependent debiasing on certification.

- Table 16 provides supplementary empirical examples for controlling the clean-perturbed performance trade-off.

*Table 6.* Measuring impact of input-dependent debiasing across target tasks for models with format perturbations (top section) and text perturbations (bottom section). Extended table of Table 1. Here red and blue indicate decrease and increase in performance/robustness due to debiasing, respectively.

| Model | Task | $\delta_{\text{drop}}$ | $\delta_{\text{clean}}$ | $\delta_{\text{pert}}$ |
|---|---|---|---|---|
| Llama 8B | 65 | -20.9 | -1.9 | 16.9 |
| Llama 8B | 69 | -19.3 | -0.8 | 14.4 |
| Llama 8B | 1297 | -55.0 | -4.0 | 30.6 |
| Llama 8B | 280 | -2.3 | -0.8 | -3.9 |
| Llama 8B | 286 | -4.7 | -4.3 | 0.3 |
| Llama 8B | 296 | -35.1 | -14.3 | 25.1 |
| Qwen 8B | 65 | -5.7 | -2.3 | 5.4 |
| Qwen 8B | 69 | -14.3 | -2.6 | 13.0 |
| Qwen 8B | 1297 | -35.6 | 0 | 36.1 |
| Qwen 8B | 220 | -6.3 | 0 | 6.0 |
| Qwen 8B | 296 | -23.3 | -8.6 | 19.8 |
| Olmo 7B | 65 | -13.4 | -9.6 | -1.3 |
| Olmo 7B | 69 | -13.3 | 0.5 | 7.3 |
| Olmo 7B | 1297 | -45.3 | 0 | 41.3 |
| Olmo 7B | 158 | -4.0 | -1.2 | 3.3 |
| Olmo 7B | 220 | -18.9 | 0 | 18.6 |
| Olmo 7B | 280 | -11.2 | 2.7 | 9.1 |
| Olmo 7B | 296 | -14.6 | -17.1 | 3.4 |
| Olmo 7B | 322 | -2.4 | -1.5 | 2.9 |
| Llama 8B | 65 | -16.7 | -9.5 | 12.6 |
| Llama 8B | 69 | -13.3 | -1.1 | 11.7 |
| Llama 8B | 1297 | -23.2 | -36.3 | 2.3 |
| Llama 8B | 280 | -18.2 | -27.8 | 10.0 |
| Llama 8B | 286 | -14.8 | 0 | 11.4 |
| Llama 8B | 296 | -19.8 | -5.7 | 18.3 |
| Olmo 7B | 65 | -21.8 | -12.9 | 5.7 |
| Olmo 7B | 69 | -19.6 | -10.9 | 2.6 |
| Olmo 7B | 1284 | -19.9 | -4.6 | 14.2 |
| Olmo 7B | 1297 | -40.6 | -24.7 | 20.9 |
| Olmo 7B | 158 | -10.4 | -4.5 | 4.6 |
| Olmo 7B | 280 | -34.0 | -26.3 | 14.7 |
| Olmo 7B | 322 | -14.8 | 3.9 | 15.9 |
| Qwen 8B | 65 | -6.1 | -1.1 | 4.9 |
| Qwen 8B | 1297 | 0.5 | -1.8 | 0.1 |
| Qwen 8B | 220 | -1.7 | 0 | 1.8 |

*Table 7.* Measuring impact of input-dependent debiasing across target tasks for models with format perturbations and model Llama 8B. Task 65, 161, 155 and other tasks demonstrate the four possible scenarios (shown in Figure 2) due to perturbation-induced bias.

| Model | Task | $\delta_{\text{drop}}$ | $\delta_{\text{clean}}$ | $\delta_{\text{pert}}$ |
|---|---|---|---|---|
| Llama 8B | 65 | -20.9 | -1.9 | 16.9 |
| Llama 8B | 69 | -19.3 | -0.8 | 14.4 |
| Llama 8B | 1283 | 1.3 | -6.5 | -6.2 |
| Llama 8B | 1284 | 5.1 | 0.4 | -4.4 |
| Llama 8B | 1297 | -55.0 | -4.0 | 30.6 |
| Llama 8B | 1347 | -2.6 | 3.9 | 6.2 |
| Llama 8B | 1419 | -8.1 | -1 | 2.6 |
| Llama 8B | 1420 | 4.4 | 8.4 | 1.5 |
| Llama 8B | 1421 | -7.0 | -6.7 | 0.5 |
| Llama 8B | 1423 | -7.7 | -10.5 | 0.5 |
| Llama 8B | 155 | 4.2 | 2.5 | -1.2 |
| Llama 8B | 158 | 0.5 | 0.7 | 1.3 |
| Llama 8B | 161 | -2.5 | -6.0 | -3.7 |
| Llama 8B | 162 | 1.3 | 2.7 | 0.4 |
| Llama 8B | 163 | -3.2 | -28.9 | -24.6 |
| Llama 8B | 1678 | -0.6 | 3.9 | 1.8 |
| Llama 8B | 220 | -3.8 | -6.1 | -0.6 |
| Llama 8B | 279 | -0.6 | 6.4 | 4.7 |
| Llama 8B | 280 | -2.3 | -0.8 | -3.9 |
| Llama 8B | 286 | -4.7 | -4.3 | 0.3 |
| Llama 8B | 296 | -35.1 | -14.3 | 25.1 |
| Llama 8B | 317 | -3.9 | -2.0 | 4.3 |
| Llama 8B | 322 | 4.7 | -1.4 | -6.1 |
| Llama 8B | 326 | 1.9 | 1.0 | -1.9 |

*Table 8.* Measuring impact of input-dependent debiasing across target tasks for models with format perturbations (top section) and text perturbations (bottom section). Here red and blue indicate decrease and increase in performance/robustness due to debiasing, respectively. This table provides the task-level information for the results shown in Table 2.

| Model | Task | $p_{\text{damage}}$ | $p_{\text{recover}}$ | $p_{\text{clean}}$ | $p_{\text{combined}}$ | $p_{\epsilon V}$ |
|---|---|---|---|---|---|---|
| Llama 8B | 65 | 26.0 | 80.9 | -2.4 | 27.8 | 81.9 |
| Llama 8B | 69 | 24.0 | 74.6 | -1.0 | 23.0 | 71.9 |
| Llama 8B | 1297 | 62.2 | 55.7 | -4.6 | 88.1 | 78.7 |
| Llama 8B | 280 | 3.2 | -169.6 | -1.1 | -5.5 | 94.6 |
| Llama 8B | 286 | 6.1 | 7.4 | -5.6 | 0.4 | 94.7 |
| Llama 8B | 296 | 36.1 | 71.6 | -14.7 | 39.2 | 80.8 |
| Qwen 8B | 65 | 6.6 | 94.5 | -2.6 | 6.5 | 51.5 |
| Qwen 8B | 69 | 15.8 | 90.7 | -2.9 | 16.7 | 63.9 |
| Qwen 8B | 1297 | 36.6 | 101.5 | 0 | 57.4 | 90.2 |
| Qwen 8B | 220 | 6.4 | 95.2 | 0 | 6.4 | 84.7 |
| Qwen 8B | 296 | 24.0 | 85.0 | -8.8 | 26.2 | 92.3 |
| Olmo 7B | 65 | 14.6 | -9.7 | -10.5 | -1.8 | 13.9 |
| Olmo 7B | 69 | 15.1 | 55.3 | 0.6 | 9.6 | 77.7 |
| Olmo 7B | 1297 | 45.3 | 91.1 | 0 | 73.7 | 71.8 |
| Olmo 7B | 158 | 8.2 | 82.7 | -2.4 | 7.2 | 78.5 |
| Olmo 7B | 220 | 19.2 | 98.0 | 0 | 22.9 | 55.6 |
| Olmo 7B | 280 | 12.3 | 81.3 | 3.0 | 11.2 | 86.0 |
| Olmo 7B | 296 | 15.7 | 23.6 | -18.5 | 4.0 | 66.3 |
| Olmo 7B | 322 | 3.1 | 120.3 | -1.9 | 3.7 | 79.2 |
| Llama 8B | 65 | 20.8 | 75.5 | -11.8 | 19.1 | 90.3 |
| Llama 8B | 69 | 16.5 | 88.5 | -1.3 | 17.0 | 87.1 |
| Llama 8B | 1297 | 26.2 | 10.0 | -41.0 | 2.4 | 39.0 |
| Llama 8B | 280 | 25.0 | 55.0 | -38.1 | 16.9 | 71.1 |
| Llama 8B | 286 | 19.2 | 77.0 | 0 | 17.8 | 88.0 |
| Llama 8B | 296 | 20.4 | 92.5 | -5.9 | 23.0 | 70.2 |
| Olmo 7B | 65 | 23.7 | 26.3 | -14.1 | 7.6 | -45.9 |
| Olmo 7B | 69 | 22.2 | 13.2 | -12.4 | 3.4 | 38.6 |
| Olmo 7B | 1284 | 24.3 | 71.6 | -5.6 | 22.3 | 85.8 |
| Olmo 7B | 1297 | 40.6 | 51.6 | -24.7 | 33.4 | 70.2 |
| Olmo 7B | 158 | 21.1 | 44.0 | -9.0 | 11.3 | 44.5 |
| Olmo 7B | 280 | 37.4 | 43.2 | -28.9 | 24.2 | 64.3 |
| Olmo 7B | 322 | 19.0 | 107.8 | 5.0 | 24.8 | 65.9 |
| Qwen 8B | 65 | 7.1 | 80.0 | -1.3 | 6.0 | 52.8 |
| Qwen 8B | 1297 | -0.5 | -20.8 | -1.8 | 0.1 | 90.1 |
| Qwen 8B | 220 | 1.7 | 106.5 | 0 | 1.8 | 91.9 |

*Table 9.* Measuring impact of input-dependent debiasing across target tasks for models with format perturbations (top section) and text perturbations (bottom section). Corresponding to Table 2 in the main text.

| | Llama 8B | | Olmo 8B | | Qwen 7B | |
|---|---|---|---|---|---|---|
| | Mean | 95% CI | Mean | 95% CI | Mean | 95% CI |
| $p_{\text{damage}}$ | 26.2 | [12.2, 44.1] | 16.7 | [11.1, 29.4] | 17.9 | [10.0, 29.9] |
| $p_{\text{recover}}$ | 20.1 | [-102.5, 65.7] | 67.8 | [34.7, 91.2] | 93.4 | [88.4, 98.1] |
| $p_{\text{clean}}$ | -4.9 | [-10.9, -2.3] | -3.7 | [-10.4, -0.2] | -2.9 | [-7.1, -0.6] |
| $p_{\text{combined}}$ | 28.8 | [8.7, 60.9] | 16.3 | [6.3, 42.8] | 22.6 | [10.4, 47.2] |
| $p_{\epsilon V}$ | 83.8 | [77.5, 90.4] | 66.1 | [41.7, 76.4] | 76.5 | [60.6, 89.2] |
| $p_{\text{damage}}$ | 21.3 | [18.8, 24.1] | 26.9 | [22.0, 34.3] | 2.8 | [0.3, 7.1] |
| $p_{\text{recover}}$ | 66.4 | [34.0, 82.8] | 51.1 | [33.7, 76.4] | 55.2 | [-20.8, 97.6] |
| $p_{\text{clean}}$ | -16.4 | [-33.7, -5.1] | -12.8 | [-20.8, -4.9] | -1.1 | [-1.7, 0.0] |
| $p_{\text{combined}}$ | 16 | [8.3, 19.6] | 18.1 | [10.4, 25.2] | 2.6 | [0.7, 6.0] |
| $p_{\epsilon V}$ | 74.3 | [54.6, 85.3] | 46.2 | [0.7, 66.7] | 78.3 | [52.8, 91.3] |

*Table 10.* Percentage of per-example of certification with/without input-dependent debiasing for target tasks with format perturbations (top section) and text perturbations (bottom section). Primed metrics are measured after debiasing. This table provides the task-level information for the per-example certification results shown in Table 3 and Figure 4.

| Model | Task | $p_C$ | $p_H$ | $p'_C$ | $p'_H$ |
|---|---|---|---|---|---|
| Llama 8B | 65 | 12.9 | 0 | 8.6 | 0 |
| Llama 8B | 69 | 0 | 0 | 4.3 | 0 |
| Llama 8B | 1297 | 0 | 0 | 0 | 0 |
| Llama 8B | 280 | 0 | 0 | 37.1 | 15.7 |
| Llama 8B | 286 | 15.7 | 0 | 51.4 | 25.7 |
| Llama 8B | 296 | 18.6 | 0 | 2.9 | 0 |
| Qwen 8B | 65 | 27.1 | 0 | 41.4 | 4.3 |
| Qwen 8B | 69 | 5.7 | 0 | 72.9 | 18.6 |
| Qwen 8B | 1297 | 0 | 0 | 2.9 | 0 |
| Qwen 8B | 220 | 0 | 0 | 94.3 | 75.7 |
| Qwen 8B | 296 | 14.3 | 0 | 67.1 | 38.6 |
| Olmo 7B | 65 | 2.9 | 0 | 7.1 | 0 |
| Olmo 7B | 69 | 0 | 0 | 7.1 | 0 |
| Olmo 7B | 1297 | 0 | 0 | 0 | 0 |
| Olmo 7B | 158 | 32.9 | 30 | 52.9 | 24.3 |
| Olmo 7B | 220 | 2.9 | 0 | 85.7 | 27.1 |
| Olmo 7B | 280 | 0 | 0 | 72.9 | 38.6 |
| Olmo 7B | 296 | 0 | 0 | 8.6 | 0 |
| Olmo 7B | 322 | 32.9 | 1.4 | 57.1 | 45.7 |
| Llama 8B | 65 | 27.1 | 0 | 20 | 1.4 |
| Llama 8B | 69 | 35.7 | 5.7 | 31.4 | 5.7 |
| Llama 8B | 1297 | 0 | 0 | 0 | 0 |
| Llama 8B | 280 | 0 | 0 | 7.1 | 0 |
| Llama 8B | 286 | 14.3 | 4.3 | 47.1 | 17.1 |
| Llama 8B | 296 | 47.1 | 18.6 | 71.4 | 12.9 |
| Olmo 7B | 65 | 0 | 0 | 8.6 | 0 |
| Olmo 7B | 69 | 28.6 | 8.6 | 10 | 0 |
| Olmo 7B | 1284 | 27.1 | 8.6 | 32.9 | 4.3 |
| Olmo 7B | 1297 | 0 | 0 | 0 | 0 |
| Olmo 7B | 158 | 0 | 0 | 2.9 | 0 |
| Olmo 7B | 280 | 0 | 0 | 4.3 | 0 |
| Olmo 7B | 322 | 5.7 | 1.4 | 35.7 | 7.1 |
| Qwen 8B | 65 | 45.7 | 15.7 | 75.7 | 21.4 |
| Qwen 8B | 1297 | 1.4 | 0 | 22.9 | 10 |
| Qwen 8B | 220 | 50 | 21.4 | 92.9 | 61.4 |

*Table 11.* Empirical verification of per-example certification: performance (BAC) on perturbed variants of certified clean examples / uncertified clean examples. Primed metrics are measured after debiasing. See Figure 5 for visualization of these results across models and tasks.

| Model | Task | $BAC_C$ | $BAC_H$ | $BAC'_C$ | $BAC'_H$ |
|---|---|---|---|---|---|
| Llama 8B | 65 | 99.7 / 58.9 | - / 59.1 | 98.9 / 73.2 | - / 75.3 |
| Llama 8B | 69 | - / 60.4 | - / 60.4 | 100.0 / 75.2 | - / 75.9 |
| Llama 8B | 1297 | - / 30.3 | - / 30.3 | - / 66.0 | - / 66.0 |
| Llama 8B | 280 | - / 69.7 | - / 69.7 | 100.0 / 34.3 | 100.0 / 64.8 |
| Llama 8B | 286 | 100.0 / 65.2 | - / 72.5 | 100.0 / 47.7 | 100.0 / 65.5 |
| Llama 8B | 296 | 100.0 / 59.7 | - / 59.8 | 97.1 / 86.8 | - / 87.1 |
| Qwen 8B | 65 | 100.0 / 75.1 | - / 77.4 | 99.9 / 75.0 | 100.0 / 85.2 |
| Qwen 8B | 69 | 100.0 / 79.7 | - / 79.8 | 99.9 / 63.0 | 100.0 / 86.8 |
| Qwen 8B | 1297 | - / 61.5 | - / 61.5 | 100.0 / 97.8 | - / 97.8 |
| Qwen 8B | 220 | - / 92.0 | - / 92.0 | 100.0 / 67.9 | 100.0 / 92.4 |
| Qwen 8B | 296 | 100.0 / 70.2 | - / 70.9 | 100.0 / 90.1 | 100.0 / 93.4 |
| Olmo 7B | 65 | 100.0 / 76.7 | - / 76.9 | 91.9 / 73.4 | - / 74.8 |
| Olmo 7B | 69 | - / 74.8 | - / 74.8 | 100.0 / 81.1 | - / 82.3 |
| Olmo 7B | 1297 | - / 53.3 | - / 53.3 | - / 95.6 | - / 95.6 |
| Olmo 7B | 158 | 100.0 / 44.6 | 100.0 / 44.6 | 100.0 / 38.4 | 100.0 / 47.3 |
| Olmo 7B | 220 | 100.0 / 84.2 | - / 84.3 | 99.5 / 85.8 | 100.0 / 96.7 |
| Olmo 7B | 280 | - / 78.3 | - / 78.3 | 100.0 / 83.0 | 100.0 / 87.5 |
| Olmo 7B | 296 | - / 79.8 | - / 79.8 | 97.1 / 80.7 | - / 81.3 |
| Olmo 7B | 322 | 100.0 / 70.3 | 100.0 / 76.8 | 100.0 / 68.4 | 100.0 / 73.7 |
| Llama 8B | 65 | 98.4 / 57.2 | - / 61.9 | 99.8 / 70.1 | 100.0 / 75.5 |
| Llama 8B | 69 | 99.7 / 63.1 | 100.0 / 67.4 | 100.0 / 69.9 | 100.0 / 77.3 |
| Llama 8B | 1297 | - / 64.7 | - / 64.7 | - / 67.2 | - / 67.2 |
| Llama 8B | 280 | - / 54.1 | - / 54.1 | 100.0 / 66.2 | - / 66.2 |
| Llama 8B | 286 | 99.6 / 56.5 | 100.0 / 60.2 | 100.0 / 54.5 | 100.0 / 68.9 |
| Llama 8B | 296 | 100.0 / 75.7 | 100.0 / 75.7 | 99.7 / 84.6 | 100.0 / 94.6 |
| Olmo 7B | 65 | - / 69.3 | - / 69.3 | 96.6 / 73.7 | - / 75.7 |
| Olmo 7B | 69 | 98.7 / 66.3 | 98.7 / 68.0 | 98.5 / 69.7 | - / 72.2 |
| Olmo 7B | 1284 | 99.0 / 56.2 | 98.7 / 60.0 | 99.5 / 61.1 | 100.0 / 74.2 |
| Olmo 7B | 1297 | - / 59.9 | - / 59.9 | - / 83.8 | - / 83.8 |
| Olmo 7B | 158 | - / 39.9 | - / 39.9 | 100.0 / 43.2 | - / 43.4 |
| Olmo 7B | 280 | - / 56.0 | - / 56.0 | 100.0 / 71.5 | - / 71.6 |
| Olmo 7B | 322 | 99.0 / 61.6 | 100.0 / 62.5 | 100.0 / 75.2 | 100.0 / 78.4 |
| Qwen 8B | 65 | 100.0 / 74.1 | 100.0 / 81.9 | 99.4 / 31.9 | 100.0 / 80.6 |
| Qwen 8B | 1297 | 100.0 / 97.5 | - / 97.5 | 100.0 / 96.0 | 100.0 / 97.5 |
| Qwen 8B | 220 | 100.0 / 94.4 | 100.0 / 96.1 | 99.9 / 61.1 | 100.0 / 86.1 |

*Table 12.* Population-level guarantees with/without input-dependent debiasing for target tasks with format perturbations (top section) text perturbations (bottom section). Primed metrics are measured after debiasing. This table provides the task-level information for the per-example certification results shown in Table 3 and Figure 4.

| Model | Task | $P_C$ | $P_H$ | $BAC_C$ | $P'_C$ | $P'_H$ | $BAC'_C$ |
|---|---|---|---|---|---|---|---|
| Llama 8B | 65 | 0 | 0 | 59.0 | 0 | 0 | 69.9 |
| Llama 8B | 69 | 0 | 0 | 59.8 | 0 | 0 | 75.2 |
| Llama 8B | 1297 | 0 | 0 | 28.1 | 0 | 0 | 72.8 |
| Llama 8B | 280 | 0 | 0 | 73.4 | 18.8 | 0 | 74.3 |
| Llama 8B | 286 | 0 | 0 | 56.2 | 31.7 | 8.5 | 56.2 |
| Llama 8B | 296 | 2.1 | 0 | 61.4 | 0 | 0 | 95.7 |
| Qwen 8B | 65 | 9.8 | 0 | 80.6 | 22.7 | 0 | 86.3 |
| Qwen 8B | 69 | 0 | 0 | 66.8 | 51.0 | 2.1 | 96.2 |
| Qwen 8B | 1297 | 0 | 0 | 52.5 | 0 | 0 | 91.8 |
| Qwen 8B | 220 | 0 | 0 | 92.4 | 70.2 | 53.5 | 100 |
| Qwen 8B | 296 | 0 | 0 | 66.1 | 45.8 | 20.1 | 96.7 |
| Olmo 7B | 65 | 0 | 0 | 78.3 | 0 | 0 | 77.5 |
| Olmo 7B | 69 | 0 | 0 | 79.5 | 0 | 0 | 89.3 |
| Olmo 7B | 1297 | 0 | 0 | 42.5 | 0 | 0 | 93.3 |
| Olmo 7B | 158 | 15.0 | 12.4 | 62.0 | 33.0 | 7.2 | 62.4 |
| Olmo 7B | 220 | 0 | 0 | 76.5 | 62.5 | 9.8 | 99.8 |
| Olmo 7B | 280 | 0 | 0 | 88.1 | 51.0 | 20.1 | 93.9 |
| Olmo 7B | 296 | 0 | 0 | 85.2 | 0 | 0 | 86 |
| Olmo 7B | 322 | 15.0 | 0 | 74.0 | 36.8 | 26.5 | 90 |
| Llama 8B | 65 | 9.8 | 0 | 54.7 | 3.4 | 0 | 60.0 |
| Llama 8B | 69 | 17.5 | 0 | 62.8 | 13.7 | 0 | 69.7 |
| Llama 8B | 1297 | 0 | 0 | 59.0 | 0 | 0 | 73.6 |
| Llama 8B | 280 | 0 | 0 | 57.3 | 0 | 0 | 69.9 |
| Llama 8B | 286 | 0 | 0 | 56.6 | 27.8 | 0.8 | 53.0 |
| Llama 8B | 296 | 27.8 | 2.1 | 74.5 | 49.7 | 0 | 92.5 |
| Olmo 7B | 65 | 0 | 0 | 65.7 | 0 | 0 | 64.9 |
| Olmo 7B | 69 | 11.1 | 0 | 73.4 | 0 | 0 | 78.3 |
| Olmo 7B | 1284 | 9.8 | 0 | 63.7 | 15.0 | 0 | 88.3 |
| Olmo 7B | 1297 | 0 | 0 | 53.4 | 0 | 0 | 79.9 |
| Olmo 7B | 158 | 0 | 0 | 46.9 | 0 | 0 | 50.8 |
| Olmo 7B | 280 | 0 | 0 | 53.8 | 0 | 0 | 70.3 |
| Olmo 7B | 322 | 0 | 0 | 63.3 | 17.5 | 0 | 87.5 |
| Qwen 8B | 65 | 26.5 | 0 | 76.1 | 53.5 | 4.7 | 82.3 |
| Qwen 8B | 1297 | 0 | 0 | 92.4 | 6.0 | 0 | 90.4 |
| Qwen 8B | 220 | 30.4 | 4.7 | 95.2 | 69.0 | 40.7 | 100 |

*Table 14.* Certification metrics at both the per-example and population levels with/without input-dependent debiasing for target tasks with format perturbations (top section) and text perturbations (bottom section). $p_C$, $p_H$ are per-example certification rates; $P_C$ and $P_H$ are population-level certification rates. Primed metrics are measured after debiasing. Corresponding to Table 3 in the main text.

| Metric | Llama 8B Mean | 95% CI | Olmo 7B Mean | 95% CI | Qwen 8B Mean | 95% CI |
|---|---|---|---|---|---|---|
| $p_C$ | 7.9 | [2.1, 14.3] | 8.9 | [1.1, 21.2] | 9.4 | [2.3, 21.7] |
| $p_H$ | 0 | - | 3.9 | [0.0, 18.8] | 0 | - |
| $p'_C$ | 17.4 | [4.0, 37.1] | 36.4 | [14.6, 58.8] | 55.7 | [21.1, 78.3] |
| $p'_H$ | 6.9 | [0.0, 17.4] | 17 | [5.7, 30.0] | 27.4 | [8.3, 56.9] |
| $P_C$ | 0.4 | [0.0, 1.4] | 3.7 | [0.0, 9.4] | 2 | [0.0, 7.9] |
| $P_H$ | 0 | - | 1.5 | [0.0, 6.2] | 0 | - |
| $P'_C$ | 8.4 | [0.0, 22.1] | 22.9 | [7.8, 41.7] | 37.9 | [14.7, 57.6] |
| $P'_H$ | 1.4 | [0.0, 5.7] | 8 | [2.5, 16.6] | 15.1 | [1.3, 42.8] |
| $p_C$ | 20.7 | [7.9, 35.5] | 8.8 | [0.8, 20.4] | 32.4 | [1.4, 48.6] |
| $p_H$ | 4.8 | [1.0, 13.1] | 2.7 | [0.2, 6.3] | 12.4 | [0.0, 19.5] |
| $p'_C$ | 29.5 | [12.9, 52.1] | 13.5 | [5.1, 26.3] | 63.8 | [22.9, 87.1] |
| $p'_H$ | 6.2 | [1.7, 12.6] | 1.6 | [0.0, 4.3] | 31 | [13.8, 61.4] |
| $P_C$ | 9.2 | [1.6, 18.8] | 3 | [0.0, 7.6] | 19 | [0.0, 29.1] |
| $P_H$ | 0.4 | [0.0, 1.4] | 0 | - | 1.6 | [0.0, 4.7] |
| $P'_C$ | 15.8 | [4.6, 34.7] | 4.6 | [0.0, 11.8] | 42.8 | [6.0, 63.8] |
| $P'_H$ | 0.1 | [0.0, 0.5] | 0 | - | 15.1 | [1.6, 40.7] |

*Table 13.* Per-example percentage of certification with/without input-dependent debiasing for target tasks with format perturbations (top section) and text perturbations (bottom section). See Table 10 for task-wise results.

| | Llama 8B Mean | Std | Olmo 7B Mean | Std | Qwen 8B Mean | Std |
|---|---|---|---|---|---|---|
| $p_C$ | 7.9 | 8.8 | 8.9 | 14.8 | 9.4 | 11.5 |
| $p_H$ | 0 | 0 | 3.9 | 10.5 | 0 | 0 |
| $p'_C$ | 17.4 | 21.5 | 36.4 | 34.4 | 55.7 | 35 |
| $p'_H$ | 6.9 | 11.2 | 17 | 19.3 | 27.4 | 30.9 |
| $p_C$ | 20.7 | 19.3 | 8.8 | 13.2 | 32.4 | 26.9 |
| $p_H$ | 4.8 | 7.2 | 2.7 | 4.1 | 12.4 | 11.1 |
| $p'_C$ | 29.5 | 26.6 | 13.5 | 14.6 | 63.8 | 36.5 |
| $p'_H$ | 6.2 | 7.3 | 1.6 | 2.9 | 31 | 27 |

*Table 15.* Population-level guarantees with/without input-dependent debiasing for target tasks with format perturbations (top section ) and text perturbations (bottom section). See Table 12 for task-wise results.

| | Llama 8B Mean | Std | Olmo 7B Mean | Std | Qwen 8B Mean | Std |
|---|---|---|---|---|---|---|
| $P_C$ | 0.4 | 0.9 | 3.7 | 6.9 | 2 | 4.4 |
| $P_H$ | 0 | 0 | 1.5 | 4.4 | 0 | 0 |
| $P_{C'}$ | 8.4 | 13.7 | 22.9 | 26.1 | 37.9 | 27.1 |
| $P_{H'}$ | 1.4 | 3.5 | 8 | 10.4 | 15.1 | 23.1 |
| $P_C$ | 9.2 | 11.6 | 3 | 5.1 | 19 | 16.5 |
| $P_H$ | 0.4 | 0.9 | 0 | 0 | 1.6 | 2.7 |
| $P'_C$ | 15.8 | 19.7 | 4.6 | 8 | 42.8 | 32.8 |
| $P'_H$ | 0.1 | 0.3 | 0 | 0 | 15.1 | 22.3 |

| $\alpha$ | $\delta_{\text{clean}}$ | $\delta_{\text{pert}}$ | $p_C$ | $p_H$ |
|---|---|---|---|---|
| *Task 065* | | | | |
| **base** | **0.0** | 0.0 | 27.1 | 0.0 |
| -0.1 | **0.0** | 4.5 | 61.4 | 17.1 |
| -0.05 | **0.0** | 6.2 | 58.6 | 11.4 |
| -0.01 | -1.4 | 5.8 | 41.4 | 4.3 |
| -0.005 | -5.6 | 5.8 | 41.4 | 4.3 |
| **0** | -2.8 | 5.8 | 41.4 | 4.3 |
| 0.005 | -2.8 | 5.8 | 41.4 | 4.3 |
| 0.01 | -2.8 | 5.8 | 41.4 | 5.7 |
| 0.05 | -2.8 | 5.9 | 41.4 | 4.3 |
| 0.1 | -2.8 | 5.9 | 41.4 | 5.7 |
| *Task 069* | | | | |
| **base** | **0.0** | 0.0 | 5.7 | 0.0 |
| -0.1 | -3.8 | 29.3 | 71.4 | 15.7 |
| -0.05 | -3.8 | 29.3 | 71.4 | 21.4 |
| -0.01 | -3.8 | 29.3 | 72.9 | 18.6 |
| -0.005 | -3.8 | 29.3 | 72.9 | 18.6 |
| **0** | -3.8 | 29.3 | 72.9 | 18.6 |
| 0.005 | -3.8 | 29.3 | 72.9 | 17.1 |
| 0.01 | -3.8 | 29.3 | 72.9 | 17.1 |
| 0.05 | **0.0** | 29.3 | 72.9 | 17.1 |
| 0.1 | **0.0** | 29.3 | 72.9 | 17.1 |
| *Task 1297* | | | | |
| **base** | **0.0** | 0.0 | 0.0 | 0.0 |
| -0.1 | **2.5** | 39.1 | 0.0 | 0.0 |
| -0.05 | **2.5** | 39.4 | 10.0 | 1.4 |
| -0.01 | **2.5** | 39.2 | 2.9 | 0.0 |
| -0.005 | **2.5** | 39.2 | 2.9 | 0.0 |
| **0** | **2.5** | 39.2 | 2.9 | 0.0 |
| 0.005 | **2.5** | 39.2 | 2.9 | 0.0 |
| 0.01 | **2.5** | 39.2 | 2.9 | 0.0 |
| 0.05 | **2.5** | 39.2 | 2.9 | 0.0 |
| 0.1 | **2.5** | 39.2 | 2.9 | 0.0 |
| *Task 296* | | | | |
| **base** | **0.0** | 0.0 | 14.3 | 0.0 |
| -0.1 | **6.7** | 29.2 | 61.4 | 32.9 |
| -0.05 | **10.0** | 30.6 | 70.0 | 34.3 |
| -0.01 | -33.3 | 30.8 | 65.7 | 40.0 |
| -0.005 | -10.0 | 30.6 | 67.1 | 40.0 |
| **0** | **3.3** | 30.6 | 67.1 | 38.6 |
| 0.005 | **3.3** | 30.6 | 67.1 | 37.1 |
| 0.01 | **3.3** | 30.6 | 67.1 | 37.1 |
| 0.05 | **6.7** | 30.6 | 67.1 | 40.0 |
| 0.1 | **6.7** | 30.6 | 67.1 | 40.0 |

*Table 16.* Effect of Gram penalty for clean sample. $\alpha$ is the parameter controlling the magnitude of the Gram penalty, enabling control over the clean-perturbed trade-off. Bold number indicate positive or null impact on clean sample performance. Measured for format perturbations on hold-out set.

