# OpenReview forum: "Harnessing Non-Adversarial Robustness in Large Language Models"
_ICML.cc/2026/Conference — ICML 2026 spotlight_

### Official Review · Reviewer_Cy3r · 2026-03-08

**Soundness:** 3
**Presentation:** 3
**Significance:** 3
**Originality:** 3
**Overall Recommendation:** 5
**Confidence:** 3

**Summary:**

This paper addresses non-adversarial robustness of LLMs to semantically-neutral prompt perturbations. The core theoretical contribution identifies perturbation-induced bias, which is a systematic expected shift in neural network module outputs under random perturbations. Motivated by this, the authors propose two debiasing methods: (1) input-independent debiasing via constant bias correction on logit neurons, and (2) input-dependent debiasing via linear ridge regression on module outputs.

**Compliance With Llm Reviewing Policy:**

Affirmed.

**Final Justification:**

Fully resolved - My concerns have been adequately addressed

**Key Questions For Authors:**

Please kindly refer to the weaknesses above.

**Limitations:**

yes

**Strengths And Weaknesses:**

**Strengths:**

- **Strong motivation grounding.** The perturbation-induced bias concept is well-motivated and clearly formalized. The unified robustness radius expressions sucessfully  link Lipschitz constants, sparsity, and bias.
- **Practical efficiency.** Both debiasing methods are closed-form and lightweight. no retraining, no multiple inference passes, no ground-truth labels are required for the unsupervised variant.
- **Principled theory-to-method pipeline.** The paper presents a clear logic: identify the bias term theoretically, and then derive improved certificates.

**Weaknesses:**

- **Narrow experimental scope.** All experiments are restricted to first-token binary/multi-class classification on Natural Instructions. The paper acknowledges this but does not provide even preliminary evidence on generation tasks, QA, or summarization. For an ICML venue, this limits perceived generality.
- **Clean-performance tradeoff is non-trivial.** Tables 1–2 consistently show negative $\delta_{clean}$. The paper frames this as acceptable when perturbations are frequent, but many deployment scenarios require high clean accuracy. No mechanism is proposed to control this tradeoff explicitly.
- **Limited baselines.** The main comparisons are against unmodified models. Randomized smoothing, self-denoising, and template ensembling are discussed qualitatively but not benchmarked head-to-head.

---

> ### Author Rebuttal · Authors · 2026-03-30
>
> **Narrow experimental scope:** Thank you for pointing this out, please see our response to **Reviewer 3AiA** for more details on the same subject "Generalization beyond classification (question 1)".
>
> **Controlling the clean-perturbed trade-off (question 2):** We are thankful for this comment. Degradation of debiased model performance on clean data could be caused by    excessive and uncontrolled facilitation of "bad or irrelevant" features through debiasing or through corrupting or even overwriting the impact of "relevant" attributes as a consequence of debiasing. Both causes could be controlled by via a  penalty term $\alpha\mathbf{\Psi}_c^{\top}\mathbf{\Psi}_c$, where $\mathbf{\Psi}_c$ contains stacked $\psi(x)=\left[x^{\top}, 1\right]^{\top}$ and $x$ are features of clean (unperturbed) examples, to the regression cost function:
>
> $$
> \\min_W\\|\\boldsymbol{\\Psi} W-\\mathbf{F}\\|^2+\\lambda\\|W\\|^2 + \\alpha \\|\\mathbf{\Psi}_c W\\|^2
> $$
>
> The corresponding solution (provided that the matrix $\\left(\\mathbf{\\Psi}^{\\top} \\mathbf{\\Psi}+\\alpha \\mathbf{\\Psi}_c^{\\top} \\mathbf{\\Psi}_c+\\lambda I\_{n+2} \\right)$ is non-singular) is:
> $$
> W=\\left(\\mathbf{\\Psi}^{\\top} \\mathbf{\\Psi}+\\alpha \\mathbf{\\Psi}_c^{\\top} \\mathbf{\\Psi}_c+\\lambda I\_{n+2} \\right)^{-1} \\mathbf{\\Psi}^{\\top} \\mathbf{F}
> $$
>
> Introduction of the extra penalty term allows balancing between reducing projection of the weights on  clean features (for $\alpha>0$) and facilitating exploitation of "clean" features (for $\alpha<0$) when other features' impact on clean performance is damaging.
> Parameter $\alpha$, therefore, enables controlling  the magnitude of the penalty and the clean-perturbed trade-off, as shown in the following table: an adequate choice of $\alpha$ can yield high $\delta_{\text{pert}}$ while retaining 0 or even positive $\delta_{\text{clean}}$ (performance improvemed in both clean and perturbed examples - see the tables below).
>
> **Table:** Regulating the impact of debiasing on clean samples' performance. $\\alpha$ is the parameter controlling the magnitude and direction of the penalty, enabling control over the clean-perturbed trade-off.
> |$\\alpha$|Task 065 $\delta_\text{clean}$|$\delta_\text{pert}$|$p_C$|$p_H$|Task 069 $\delta_\text{clean}$|$\delta_\text{pert}$|$p_C$|$p_H$|Task 1297 $\delta_\text{clean}$|$\delta_\text{pert}$|$p_C$|$p_H$|Task 296 $\delta_\text{clean}$|$\delta_\text{pert}$|$p_C$|$p_H$|
> |:----|--------:|-------:|--------:|-----:|---------:|----:|---------:|---------:|-------:|------:|----:|-----:|-------:|------:|----:|-----:|
> | **base** |**0**|  0|27.1 | 0|**0**|  0| 5.7 | 0| **0**|0|  0|  0|**0**|  0|14.3 | 0|
> | -0.1 |**0**|  4.5 |61.4 |17.1 |  -3.8 | 29.3 |71.4 |15.7 | **2.5** |  39.1 |  0|  0|**6.7** | 29.2 |61.4 |32.9 |
> | -0.05|**0**|  6.2 |58.6 |11.4 |  -3.8 | 29.3 |71.4 |21.4 | **2.5** |  39.4 | 10|  1.4 |  **10**| 30.6 |70|34.3 |
> | -0.01|  -1.4 |  5.8 |41.4 | 4.3 |  -3.8 | 29.3 |72.9 |18.6 | **2.5** |  39.2 |  2.9 |  0| -33.3 | 30.8 |65.7 |40|
> | -0.005  |  -5.6 |  5.8 |41.4 | 4.3 |  -3.8 | 29.3 |72.9 |18.6 | **2.5** |  39.2 |  2.9 |  0| -10| 30.6 |67.1 |40|
> | **0** |  -2.8 |  5.8 |41.4 | 4.3 |  -3.8 | 29.3 |72.9 |18.6 | **2.5** |  39.2 |  2.9 |  0|**3.3** | 30.6 |67.1 |38.6 |
> | 0.005|  -2.8 |  5.8 |41.4 | 4.3 |  -3.8 | 29.3 |72.9 |17.1 | **2.5** |  39.2 |  2.9 |  0|**3.3** | 30.6 |67.1 |37.1 |
> | 0.01 |  -2.8 |  5.8 |41.4 | 5.7 |  -3.8 | 29.3 |72.9 |17.1 | **2.5** |  39.2 |  2.9 |  0|**3.3** | 30.6 |67.1 |37.1 |
> | 0.05 |  -2.8 |  5.9 |41.4 | 4.3 |**0**| 29.3 |72.9 |17.1 | **2.5** |  39.2 |  2.9 |  0|**6.7** | 30.6 |67.1 |40|
> | 0.1  |  -2.8 |  5.9 |41.4 | 5.7 |**0**| 29.3 |72.9 |17.1 | **2.5** |  39.2 |  2.9 |  0|**6.7** | 30.6 |67.1 |40|
>
> **Limited baselines:** Thank you for the comment. The reason why we did not include head-to-head comparisons is because we believe that these methods are not directly comparable, head-to-head. (1) From inference-cost perspective, randomised smoothing, self-denoising and template ensembling require multiple inference passes per single prompt. Our method needs only one pass in inference. (2) From application perspective, all the other methods change the input prompt going into the model, while we intervene with the output logits or intermediate activations of the model. (3) From the viewpoint of perturbation model and certification rates, the three methods are focused on worst case $l_2$-radius to guard against distribution-agnostic adversarial perturbations. Our focus is on typical performance on non-adversarial nominal perturbations with data-aware certificates. These differences in costs, applications and aims obstruct fair comparisons, and that is why we did not consider them as good baselines. Even though we discuss most of these in the paper, we will be happy to add a further clarifying comment on these differences in the paper and explain why direct head-to-head comparisons could be unfair (in both ways).

---

> > ### Author Rebuttal · Reviewer_Cy3r · 2026-04-01
> >
> > Fully resolved - My concerns have been adequately addressed

---

### Official Review · Reviewer_tnw2 · 2026-03-12

**Soundness:** 3
**Presentation:** 3
**Significance:** 3
**Originality:** 3
**Overall Recommendation:** 5
**Confidence:** 4

**Summary:**

This paper addresses the fragility of Large Language Models  to non-adversarial, semantically-neutral prompt perturbations such as formatting changes or minor text variations. The authors approach this problem theoretically, identifying a critical phenomenon they term perturbation-induced bias. They demonstrate that when an input prompt is perturbed, the expected center of the internal representation shifts away from the unperturbed representation, pushing the features closer to decision boundaries and eroding the model's robustness margin. To counter this, the authors propose a lightweight, training-free debiasing technique by estimating this shift from perturbed samples, they apply simple post-hoc corrections to the module outputs.

**Compliance With Llm Reviewing Policy:**

Affirmed.

**Final Justification:**

The rebuttals inclusion of generalization experiments show promise in the work. With my prior assessment being fairly confident, I do believe these experiments help the claims in the paper and are now more of interest.

**Key Questions For Authors:**

1) Can you provide the confidence intervals table 3 and table 2.
2) How well do you expect results to generalize across generative tasks.

**Limitations:**

Yes

**Strengths And Weaknesses:**

Strengths:
1) The method introduced is training-free and unsupervised which is a key plus point to the usefulness of the methodology proposed.
2) The theoretical analysis section is excellent and provides much needed formalization of the intuitions/hypotheses presented.
3) The presentation is very well done, the writing is clear and the flow of the paper is consistent. No issues here.
4) Figure 4 and Figure 5 are excellent results and worth emphasizing and exploring further.
5) Overall, the contribution is novel and experiments soundness of the work is consistent.
6) The problem studied is of interest to some community and the research problem focuses on concerns that genuine and empirically validated.

Weakness:
1) The primary concern I take with this work is that the experiments are on classification, this is the main problem with the work. Generalization to generative tasks is crucial in case of the modern LLMs, especially the ones tested in the paper.
2) Degradation on clean prompts is problematic. Further experimentation on "general coherence" benchmarks like HellaSwag, BoolQ, MMLU-Pro etc are needed to full understand the accuracy tradeoff. Without these experiments, the potential downsides of the methodology proposed are unclear at best and negatively suggestive at worst.
3) The method introduced assumes sampling perturbations from a known distribution and real world perturbations might not align with this. This setting doesn't seem particularly realistic.

Overall, although I do think that work has some experimental

---

> ### Author Rebuttal · Authors · 2026-03-30
>
> **Generalization beyond classification:** Thank you for pointing this out, please see our response to **Reviewer 3AiA** (question 1) where we demonstrated generalization beyond classification to generation and summarization tasks.
>
> **Degradation on the clean prompts:** Thank you for pointing this out, please see our response to **Reviewer Cy3r** (question 2) for a method to control clean-perturbed trade-off.
>
> In addition, the reviewer asks about "general coherence" which we understand as how debiasing in one task may affect  performance of the model on other tasks.
>
> For logit debiasing, the setting assumes that the model is purposed for one task, since we need to know which logits to debias, therefore testing on "general coherence" benchmarks are not relevant for this setting.
>
> For debiasing in the intermediate layers, "general coherence" becomes relevant. Due to time limitations, we performed a small experiment where we evaluate original models and models that have been debiased (with the method in the response to Reviewer 3AiA) on two small 300 clean example subsets of HellaSwag and BoolQ. For the model debiased on Natural Questions at layer 8 (L8), performance  degradation on HellaSwag and BoolQ was a mere -0.67\% and -2.33\%, whilst recovering 105.5\% of the 9.5\% Drop in EM for Natural Questions due to perturbations. For the model debiased on the TriviaQA dataset at L8, the performance degradation on HellaSwag and BoolQ was -1.33\% and -2\%, whist recovering 83.4\% of a 35.8\% drop in EM. Though the performance degradation is small, it is worth noting in the paper and worth further exploration. We will gladly include these results in our revised paper.
>
> **Perturbation distributions:** We appreciate the concern. From theoretical viewpoint, assuming that data used in training and testing phases are sampled from the same distribution is standard. In reality, this assumption may not hold true. In our response to the previous question, we hope that we demonstrated that the method has a degree of robustness to distribution shifts and changes of tasks. As for the information inferred from training samples, the method infers very limited information from samples of perturbations (see the  introduction of the box B in Appendix D), and this information (bounds of the box B) can be estimated empirically from real world data consistent with conditions of model's deployment.
>
> **Confidence intervals for Table 3 and Table 2:** Please find in the tables below part of Table 2 and Table 3 with confidence intervals (CI), calculated as task-level 95\% CI w.r.t. mean with bias-corrected accelerated bootstrapping with 10k resamples. For each perturbation type, the majority of models (2/3) show high lower 95\% CI bounds above 30\% indicating that the recovery effect remains strong in the majority of settings, while $p_{\epsilon V}$ indicate consistent gains in robustness radius. We will be happy to include the full version of these tables in the Appendix and discuss them in the main text.
>
> **Table:** Measuring impact of input-dependent debiasing across target tasks for format perturbations. Corresponding to Table 2 in the main paper.
> | Metric     |   Llama 8B Mean |  CI   |   Qwen 8B Mean | CI   |   Olmo 7B Mean | CI    |
> |:-----|--:|:----|-----:|:--|---------------:|:--------------|
> | $p_{\text{damage}}$   | 26.2 | [13.1, 45.4]  |17.9 | [10.0, 29.9] | 16.7 | [11.0, 29.4]  |
> | $p_{\text{recover}}$  | 20.1 | [-99.4, 65.5] |93.4 | [88.1, 98.0] | 67.8 | [35.6, 90.7]  |
> | $p_{\text{clean}}$    | -4.9 | [-10.9, -2.3] |-2.9 | [-6.5, -0.6] | -3.7 | [-10.5, -0.2] |
> | $p_{\text{combined}}$ | 28.8 | [10.5, 61.6]  |22.6 | [10.4, 44.9] | 16.3 | [6.4, 44.2]   |
> | $p_{\epsilon V}$      | 83.8 | [77.5, 90.4]  |76.5 | [59.6, 88.8] | 66.1 | [41.8, 76.3]  |
>
> **Table:** Certification metrics at both the per-example and population levels with/without input-dependent debiasing for target tasks with format perturbations. $p_C$, $p_H$ are per-example certification rates; $P_C$ and $P_H$ are population-level certification rates. Primed metrics are measured after debiasing. Corresponding to Table 3 (bottom section) in the main body.
>
> | Metric     |   Llama 8B Mean |  CI   |   Qwen 8B Mean | CI   |   Olmo 7B Mean | CI    |
> |:---------|----------------:|:--------------|---------------:|:-------------|---------------:|:-------------|
> | $p_{C}$  |  7.9 | [2.1, 14.3]   | 9.4 | [2.3, 20.3]  | 8.9 | [1.1, 21.2]  |
> | $p_{H}$  |  0   | -    | 0   | -  | 3.9 | [0.0, 15.4]  |
> | $p_{C}'$ | 17.4 | [4.0, 38.3]   | 55.7 | [21.1, 78.3] | 36.4 | [15.7, 60.4] |
> | $p_{H}'$ |  6.9 | [0.0, 18.1]   | 27.4 | [8.3, 60.6]  |17   | [5.7, 30.7]  |
> | $P_C$    |  0.4 | [0.0, 1.4]    | 2   | [0.0, 7.9]   | 3.7 | [0.0, 9.4]   |
> | $P_H$    |  0   | -   | 0   | -  | 1.5 | [0.0, 7.7]   |
> | $P_{C}'$ |  8.4 | [0.0, 21.1]   | 37.9 | [14.7, 57.6] | 22.9 | [7.8, 40.8]  |
> | $P_{H}'$ |  1.4 | [0.0, 5.7]    | 15.1 | [1.3, 42.8]  | 8   | [2.5, 16.2]  |

---

> > ### Author Rebuttal · Reviewer_tnw2 · 2026-04-03
> >
> > Thank you for the response. My concerns are resolved. The generalization experiments are a good addition. I will be updating my score to reflect these changes.

---

### Official Review · Reviewer_TWqC · 2026-03-12

**Soundness:** 3
**Presentation:** 3
**Significance:** 2
**Originality:** 3
**Overall Recommendation:** 4
**Confidence:** 4

**Summary:**

This paper addresses the critical challenge of improving Large Language Models' (LLMs) robustness to semantically neutral prompt perturbations without expensive full-model retraining. The authors identify perturbation-induced bias as a key factor undermining robustness. Through theoretical analysis, they establish formal robustness certificates linking traditional determinants (Lipschitz constants, margins) with this newly identified bias. The core contribution is a computationally efficient debiasing method that compensates for this bias through simple parameter adjustments, requiring neither full retraining nor extensive supervisory data. Extensive experiments across multiple models (Llama, Qwen, Olmo) and perturbation types demonstrate significant recovery of performance degradation (up to 35% improvement) while providing provable robustness guarantees.

**Compliance With Llm Reviewing Policy:**

Affirmed.

**Final Justification:**

My concerns have been addressed, so I am maintaining my positive score.

**Key Questions For Authors:**

1. Could you provide concrete examples of "semantically similar but textually different prompts" that motivated this research?
2. Have you tested the method on unseen perturbations?

**Limitations:**

yes

**Strengths And Weaknesses:**

Strengths

1. The paper makes a fundamental theoretical contribution by formally characterizing how random perturbations cause systematic shifts in LLM feature representations.
2. The proposed debiasing approach stands out for its simplicity and effectiveness.
3. The paper provides thorough experimental validation across 52 tasks from Natural Instructions, covering both format and text perturbations.

Weaknesses

1. The abstract prominently raises the issue of "semantically similar but textually different prompts" as a core challenge for LLM robustness, yet the paper fails to provide specific examples or detailed characterization of what constitutes such problematic prompts. This omission significantly undermines the practical relevance and contextual understanding of the research problem.
2. While the theoretical framework elegantly models perturbation-induced bias, the connection to real-world "semantically similar but textually different" scenarios remains abstract. The experimental perturbations (format changes, character-level modifications) may not adequately represent the semantic equivalence challenges mentioned in the abstract.
3. While the text mentions using Natural Instructions (Wang et al., 2022) and following the protocol of Seleznyov et al. (2025), it does not specify which individual tasks​ from that vast dataset were selected for the final experiments. The description of applying "additional task filters" is too vague to allow a reader to know precisely what problems were being tested. This lack of specificity becomes a significant issue when examining Table V. The table reports results on several methods, but without a clear list of the underlying tasks, the numbers are presented in a vacuum.

---

> ### Author Rebuttal · Authors · 2026-03-30
>
> **Examples of semantically similar but textually different prompts:** We are thankful for the query. In the revised text we will add examples of both format and text perturbations. Please find below a random example from Task 065 with two concise format perturbations and two text perturbations. We will also clarify that "semantically similar but textually different" prompts are "prompts that are semantically equivalent to a human reader but differ in surface textual forms".
>
>
> ```
> Original
> --------
> Sentence 1: Marion and Louise had a pet parrot named Preacher...
> Sentence 5:  I said hello back, and he asked how I was doing
> Option 1: When I walked in, Preacher said hello.
> Option 2: When I walked in, Preacher said hello but will never speak again.
>
> Format Pert 1
> -------------
> Sentence	1): Marion and Louise had a pet parrot named Preacher...
> Sentence	5):  I said hello back, and he asked how I was doing,
> Option	i ) : When I walked in, Preacher said hello.
> Option	ii ) : When I walked in, Preacher said hello but will never speak again.,
>
> Format Pert 2
> -------------
> Sentence [I]- Marion and Louise had a pet parrot named Preacher. --...
> Sentence [V]-  I said hello back, and he asked how I was doing --
> Option i.	When I walked in, Preacher said hello.
> Option ii.	When I walked in, Preacher said hello but will never speak again. --
>
> Text Pert 1
> -----------
> Sentence 1: Marion and Louise had a pet parrot named Preacher...
> Sentence 5:  I\xa0said hello back, and he asked how I was doing
> Option 1: When I walked in, Preacher said hello.
> Option 2: When I walked in,\xa0Preacher said hello but will never speak again.
>
> Text Pert 2
> -----------
> Sentence 1: Marion and Louise had a pet par\u200drot named Preacher...
> Sentence 5: I\xa0said hello back, and he asked how I\xa0was doing
> Option 1: When I walked in, Preacher sai\u200cd hello.
> Option 2: When I walked in, Preacher sa\ufeffid hello but wi\u200cll never speak again.
> ```
>
> **Link between theoretical perturbations and real-world "semantically similar but textually different":** We are grateful for the opportunity to clarify this important point. Our theoretical analysis allows dealing with real perturbations with only limited information at hand. This is reflected in the introduction of a box in Appendix D, which can be empirically estimated. This setting enables linking samples from real-world "semantically similar but textually different" perturbations with theoretical certificates and guarantees. The former provides an environment to bring the analysis to application.
>
> **Tasks and task filters:** Please find below a table for individual task names and task descriptions, we will be happy to include the full table in the appendix of the paper and provide reference and summary in the main body.  The task filtering approach has been described in Section 4 Experimental Setup: (1) a >1% drop in performance due to perturbations, and (2) balanced accuracy $\geq$ 70% on the clean examples. This filtering isolates tasks for which the model is performant on clean examples  and is noticeably less accurate under semantically-neutral prompt alterations.
>
>
> | Task No. | Task Name | Task description |
> | -------- | ---------------------------------------------- | ------------------------------------------------------------------------------------------------------------------- |
> | 065  | Time-travel consistent sentence classification | Choosing the option that makes a given short story consistent                                              |
> | 069 | Abductive NLI classification    | Choosing text that completes a story based on given beginning and ending.                                           |
> | 1297  | QASC question answering  | Given two facts, and a multiple-choice question, answer the question.                                               |
> | ...      | ...  | ...            |
> | 158      | Count frequency of words  | Count the number of occurrences of a word in the given sentence.                                                    |
> | 322      | Jigsaw classification threat   | Given a comment from online platforms, classify whether or not it contains threats.                                 |
>
> **Tests on unseen perturbations:** The very motivation of our proposed method is to guarantee robust performance using minimal supervisory information. That is why testing is performed using held-out examples paired with held-out perturbations that are never seen during training. Additionally, population-level certificates are evaluated on a separate hold-out set comprising 15\% of examples and perturbations. This is mentioned in Appendix G.1, but we will further clarify and emphasize this in the main text as well.

---

> > ### Author Rebuttal · Reviewer_TWqC · 2026-04-01
> >
> > Thank you for your response. The answer addressed my concerns, so I have increased my score.

---

### Official Review · Reviewer_3AiA · 2026-03-12

**Soundness:** 3
**Presentation:** 3
**Significance:** 2
**Originality:** 3
**Overall Recommendation:** 4
**Confidence:** 3

**Summary:**

This paper addresses the robustness of LLMs to semantically-neutral prompt perturbations (e.g., formatting changes, typos, Unicode confusables). The authors identify "perturbation-induced bias"—a systematic shift in expected module outputs under random perturbations—as a critical factor behind robustness degradation. They provide theoretical robustness certificates linking Lipschitz constants, sparsity, and this bias term into unified expressions. Motivated by the theory, they propose two debiasing methods (input-independent and input-dependent) that correct for this systematic shift without full model retraining or access to ground-truth labels. Experiments on Natural Instructions tasks with three open-weight models (Qwen-3-8B, Llama-3.1-8B, Olmo-3-7B) show that debiasing can recover a significant portion of performance lost to perturbations and improve formal robustness certification rates.

**Compliance With Llm Reviewing Policy:**

Affirmed.

**Key Questions For Authors:**

1. **Generalization beyond classification:** Can you provide any preliminary evidence (even on a small scale) that the debiasing framework improves robustness in generation tasks (e.g., question answering, summarization)? This would significantly strengthen the practical relevance of the work.

2. **Controlling the clean-perturbed trade-off:** Is there a principled way to control the trade-off between clean and perturbation, e.g., via a regularization parameter? The current approach seems to accept whatever trade-off the optimal bias region produces.

3. **Tighter bounds:** Have you explored using distributional assumptions (e.g., sub-Gaussian perturbations) to derive tighter certification bounds? The current bounds are very conservative, and practical perturbations (typos, formatting) likely have more structure than worst-case bounded noise.


4. **Comparison to existing methods:** The paper positions itself against randomized smoothing and LoRA but does not directly compare debiasing + certification against randomized smoothing certification on the same tasks. A head-to-head comparison on certification rates and computational cost would be informative.

**Limitations:**

Yes, the authors have provided a thorough discussion of limitations in Section 6.3, including the restriction to first-token tasks, the clean-perturbed trade-off, conservative bounds, and the need for white-box access. The impact statement appropriately frames the positive societal implications.

**Strengths And Weaknesses:**

### Strengths

1. **Strong theoretical grounding.** The paper provides a principled analysis connecting perturbation-induced bias to robustness degradation. The theoretical certificates (Theorems B.1, B.5, F.1) unify Lipschitz constants, variance, and bias into coherent robustness bounds. The proofs are clearly presented and appear correct upon inspection.

2. **Practical and lightweight intervention.** The proposed debiasing methods are computationally efficient—input-independent debiasing requires only a scalar offset per neuron, and input-dependent debiasing uses closed-form linear ridge regression. This is a significant advantage over methods requiring full retraining or multiple inference passes (e.g., randomized smoothing).

3. **Comprehensive experimental design.** The paper evaluates across three models, two perturbation types (format and text), and multiple tasks from Natural Instructions. The metrics are well-chosen (δdrop, δpert, δclean, robustness radii, certification rates), giving a multifaceted view of the trade-offs.

4. **Theory-practice alignment.** The four theoretical cases in Figure 2 (panels a–d) are all observed empirically, which lends credibility to the theoretical framework. The connection between debiasing and improved certification (Table 3, Figure 4) is compelling.

5. **Unsupervised applicability.** The input-dependent debiasing method does not require ground-truth labels, only clean examples and perturbation samples, making it practical for deployment scenarios.

### Weaknesses

1. **Narrow task scope.** Experiments are restricted to first-token prediction in classification tasks. While the authors acknowledge this limitation (Section 6.3), the practical applicability to generation tasks, the dominant use case for LLMs remains unvalidated. The theoretical framework in principle applies more broadly, but this gap weakens the empirical contribution.

2. **Clean-performance degradation.** A recurring pattern across experiments is that debiasing improves perturbed performance at the cost of clean-example accuracy (negative $\delta$ clean in Tables 1–2). The paper does not provide a principled mechanism for controlling this trade-off, and the Pareto frontier is left for future work.

3. **Limited investigation of intermediate layers.** The current approach focuses on the logit layer. The brief mention of intermediate-layer debiasing (Section 6.2) uses a very small calibration set (m=10) and only three tasks on one model. Systematic exploration of layer selection could yield stronger results.

4. **Presentation issues.** The paper is dense, with the main body covering theory and experiments in 8 pages while deferring most proof details and many experimental results to a 12-page appendix. Some key results (e.g., per-task breakdowns in Table 7) are only in the appendix, making it hard to assess consistency across tasks from the main text alone.

---

> ### Author Rebuttal · Authors · 2026-03-30
>
> **Generalization beyond classification (question 1):** We are grateful for the opportunity to evidence applicability of our method to tasks beyond classification. The focus on classification was motivated by shortcomings inherent to Batch Calibration and Randomized Smoothing (see Related Work section).  We mentioned a possibility to apply the method beyond classification, and in hindsight we should have indeed presented relevant examples in the original submission. We would be happy to include these in the revision. Below we explain how the method can be applied to generation or summarization.
>
> Generalization to generation tasks  necessitates moving away from mere logits debiasing to debiasing features in intermediate layers, as otherwise there would be too many outputs (logits) to debias.  To achieve this we select an intermediate layer and (1) extract $\mathbb{R}^n$ feature vectors of the last token before generation for both clean and perturbed input prompts, (2) compute the differences between clean and perturbed feature vectors for the training set and their mean $\mu$, define $\phi(x) = x-\mu$, for feature vector $x$, (3) compute PCA on the training set comprising $\phi(x)$, (4) select $k$ principal components and perform the ridge regression-based bias correction on projections to these $k$ principal components.  If $H=(h_1, \dots, h_n)$ is the matrix of principal components, and $\theta:\mathbb{R}^n\rightarrow\mathbb{R}^k$ is the debiasing function (such as the ridge regression) then intermediate layer debiasing can be achieved as:
> $$
> \\phi_{\\mathrm{debiased}}(x)=H \\left(
> \\begin{array}{l}
> \\theta_1 (H^{T}\phi(x))+h_{1}^{T}\phi(x) \\\\
> \\vdots \\\\
> \\theta_k(H^{T}\\phi(x))+h_{k}^{T}\\phi(x)\\\\
> h_{k+1}^{T}\phi(x)\\\\
> \\vdots\\\\
> h_n^{T}\\phi(x)
> \\end{array}\\right).
> $$
> For $k=1$, this reduces the problem precisely to the one we analysed in the paper. In experiments we therefore set $k=1$
>  (debiasing was applied only to projections on the first principal component leaving other variables untouched).
>
> **Table:** Effect of debiasing in intermediate layers.  "Drop" measures percentage damage due to text perturbations. Percentage of the drop in performance recovered (PR or $p_\text{recover}$) is measured for debiasing at layers 0, 8, 16, 32 for Qwen-8B.
> |Task| Metric|Drop|L0 PR|L8 PR|L16 PR|L32 PR|
> |:-----|:---|:---|---:|---:|---:|---:|
> |GSM-8k|EM|34.3|63.3| 68.8 |59.4|-79.5|
> |Natural Questions|EM |9.5|104.5|105.5|79.6|-31.4|
> |TriviaQA|EM|35.9|87.5|83.4|58.4|-33.7|
> |XSum|F1|8.9|123.1|111.8|56.0|-125.9|
>
> Examples of such debiasing for generation tasks is shown in table above.  The tasks comprise small 275 example subsets of TriviaQA, GSM-8K, Natural-Questions (generation tasks) and XSum (summarisation task) with 100 text perturbations per example (train-test split: 100/175 examples and 30/70 perturbations). The method shows high recovery rate (>50% for most layers except for L32) for the drop in performance due to text perturbations. These experiments, produced within limited timeframe available for response, evidence feasibility of generalization beyond classification. We will be pleased to include them.
>
> **Controlling the clean-perturbed trade-off:** Thank you for pointing this out, please see our response to **Reviewer Cy3r** (q2) for the mechanism to control clean-perturbed trade-off.
>
> **Tighter bounds:** We appreciate the comment, but any bounded variable is sub-gaussian (Vershynin, 2018 - Page 25), and we have already used an appropriate inequality (Hoeffding) in Theorems F.1 (per-example certificates) and Remark F.4 (population-level certificates). In other places, we used the McDiarmid inequality giving sub-gaussian tail bounds. We believe the comments concerns our use of Chebyshev inequalities in Theorem B.5 and Remark F.2. Through extensive empirical assessment at time of writing the paper, we found that due to low dimensionality, Hoeffding bounds did not provide a good measure of certification in practice in the target tasks. That is why we used Chebyshev inequality which turned out to be tighter in these settings. In addition, our analysis serves to reveal the critical perturbation-induced bias and demonstrate debiasing improves certification, so our main interest lies in the relative difference in certification radii/rates, which we demonstrated tangible improvements upon.
>
> **Comparison to existing methods:** With respect to randomised smoothing, we position the paper as falling into the same category of aiming to provide robustness guarantees, but what our methods requires and what our certificate provides differs significantly from randomised smoothing. Please see our response to **Reviewer Cy3r** for comparison. With respect to LoRA, Section 6.2 stresses the need for additional supervisory information if LoRA were to be applied. In contrast, our methods from logit debiasing and new debiasing method (answer to question 1) for intermediate layers requires no supervisory information.

---

> > ### Author Rebuttal · Reviewer_3AiA · 2026-04-02
> >
> > Most of my concerns have been adequately addressed.

---

> > > ### Author Response · Authors · 2026-04-03
> > >
> > > Thank you for your review and feedback. Unfortunately, we have not found any follow-up questions or concerns in the rebuttal acknowledgement and that's why we cannot provide further clarifications.

---

### Decision · Program_Chairs · 2026-04-30

**Decision:**

Accept (spotlight)

**Comment:**

The paper explores strategies for increasing LLMs' robustness to semantically-neutral modifications to an input prompt, without the need for complete re-training. They show that a little bit of finetuning can debias models and improve robustness to input perturbations. All reviewers appreciate the theoretical grounding of the approach, and the well-executed method and experiments. Therefore, I recommend acceptance.

A small note: please edit your figures to follow the ICML accessibility guidelines---notably, that "font size in figures should be no smaller than the font size of the caption of the figure." Figure 3 is especially hard to read.